# MULTI-LEVEL CLIP KNOWLEDGE TRANSFER FOR OPEN-VOCABULARY OBJECT DETECTION

## ABSTRACT

Open-vocabulary object detection (OVOD) aims to detect novel objects beyond the training categories. Recent approaches extend conventional detectors to OV detectors by combining their detector scores with zero-shot classification scores of pre-trained vision-language models such as CLIP, which is capable of identifying various visual concepts via language descriptions. However, such a simple score-level combination struggles to balance the localization and classification of novel objects: CLIP encodes global semantics for accurate classification but exhibits limited sensitivity to localization precision when scoring proposals, whereas the detector provides robust localization yet tends to misclassify novel objects as background. Instead of a trade-off, our goal is to leverage the complementary strengths of CLIP and the detector. To this end, we propose the **M**ulti-level **CLIP T**ransfer (MCT-Det) strategy, which effectively transfers context, alignment, and generalization knowledge from CLIP to the detector at three distinct levels. Specifically, for each region proposal: 1) At the *feature-level*, we refine region features by dynamically integrating CLIP's global context via cross-attention to improve localization. 2) At the *embedding-level*, we integrate the region representations of CLIP and the detector into a unified embedding to couple image-text alignment with localization-awareness for reliable recognition. 3) At the *score-level*, we follow previous methods to exploit CLIP's zero-shot classification ability via the scores combination strategy. Building upon F-ViT, our MCT-Det achieves comprehensive improvements and outperforms state-of-the-art methods, with $52.9$ $\text{AP}_{50}^{novel}$ on OV-COCO and $39.8$ $\text{mAP}_r$ on OV-LVIS using ViT-L/14.

## 1 INTRODUCTION

Object detection is a fundamental task in computer vision that aims to localize and categorize objects within images, serving as a cornerstone for higher-level applications such as Embodied Intelligence, especially in visual navigation, object manipulation, and autonomous exploration. It has seen remarkable progress with models such as YOLO series (Redmon et al., 2016; Tian et al., 2025), R-CNN series (Girshick et al., 2014; Ren et al., 2015; He et al., 2017), and DETR series (Carion et al., 2020; Zhu et al., 2020; Zhang et al., 2022). However, due to the scarcity and high cost of large-scale annotated detection data, these detectors are typically constrained to the closed-set - the recognizable categories are predefined and fixed during both training and testing, making them ill-suited for real-world scenarios where object categories often extend beyond the scope. To address this limitation, open-vocabulary object detection (OVOD) (Zareian et al., 2021) has been formulated, which incorporates language modality as auxiliary guidance to enable recognition of novel and unseen categories, thereby significantly enhancing the practicality and scalability of detectors.

Recent advances in vision-language models, such as CLIP (Radford et al., 2021), have demonstrated impressive zero-shot recognition capabilities and provided strong support for OVOD methods. To avoid costly additional training, F-ViT (Wu et al., 2023b) directly adopts the CLIP visual encoder as a frozen backbone in a standard Mask R-CNN detector. During inference of F-ViT, the trained detector head provides detection boxes and masks, while the final scores are computed as the geometric mean of **CLIP** and **detector** scores. With its ease of training and extensibility, F-ViT has become a widely adopted base framework in the OVOD task. Recent works mainly focus on enhancing the

Figure 1: Failure case of score-level combination: CLIP assigns similar scores to poor and well-located boxes while the detector misclassifies novel objects as background. In some cases, the combined score fails to reflect the relative quality of boxes, hindering the detector's post-processing.

region-text alignment of ViT-based CLIP models (Jing et al., 2024; Li et al., 2024; Qiu et al., 2025; Wang et al., 2025b; Xie et al., 2025) for better region-level recognition, while still following the F-ViT pipeline that relies on score-level combination for open-vocabulary classification.

The score-level combination effectively exploits CLIP's zero-shot classification capability and significantly improves the performance on recognizing novel objects, however, a notable precision gap remains between novel and base categories (for F-ViT using ViT-B/16 CLIPSelf as backbone, $AP_{50}^{base}$ is 54.2 while $AP_{50}^{novel}$ is 33.6). Based on experimental analysis in Sec. 4.2, we argue that relying solely on the score-level combination is insufficient to overcome the inherent limitations of both CLIP and the detector. For each region proposal, CLIP scores provide accurate classification results yet lack sensitivity to the proposal's localization quality. In contrast, detector scores reflect the localization quality well, but tend to recognize unseen objects as background. As illustrated in Fig. 1, the score-level combination serves as a trade-off between CLIP and detector scores, and remains insufficient to integrate localization with classification, resulting in similar scores of proposals with different localization quality. To better exploit the complementary strengths and overcome the respective limitations of CLIP and the detector, it is necessary to explore integration strategies beyond the score-level.

In this paper, we propose MCT-Det, a Multi-Level CLIP Knowledge Transfer strategy that effectively leverages the complementary strengths of CLIP and the detector. For each region proposal: 1) At the ***feature-level***, we employ cross-attention-based feature interaction between the region features encoded by CLIP and the detector, which transfers CLIP's global context to enhance the region awareness for better localization. 2) At the ***embedding-level***, we establish a direct connection from CLIP's region embedding to the detector's embedding, which transfers CLIP's image-text alignment capability for better classification and further contributes to scoring proposals. We also unify the regression and classification heads to better couple localization and recognition. 3) At the ***score-level***, we follow previous methods to combine CLIP and detector scores through geometric mean, which transfers CLIP's zero-shot classification results to the OVOD task. The proposed feature- and embedding-level transfers, together with the existing score-level transfer, form a multi-level CLIP knowledge transfer strategy that effectively leverages CLIP for the OVOD task and leads to significant improvements. Meanwhile, our approach is compatible with recent advances in enhancing CLIP's region-awareness, allowing for further gains with stronger backbones.

Our observations and contributions can be summarized as follows:

- We experimentally analyze the F-ViT and reveal that the score-level combination is inadequate to overcome CLIP's insufficient localization capability and the detector's limited classification capability when detecting novel objects.
- We propose MCT-Det, which builds upon F-ViT and leverages the complementary strengths of CLIP and the detector through feature-, embedding-, and score-level transfer strategies.
- Extensive experiments demonstrate that MCT-Det significantly improves the detection performance on novel categories and surpasses state-of-the-art methods, achieving 52.9 (+8.5 gain over baseline) $AP_{50}^{novel}$ on OV-COCO and 39.8 (+4.9 gain) $mAP_r$ on OV-LVIS using ViT-L/14 as the backbone.

## 2 RELATED WORK

**Close-set object detection.** Close-set object detectors are limited to a pre-defined set of categories, meaning they can only recognize objects encountered during the training phase. These detectors are generally categorized into region-based, pixel-based, and query-based models. Region-based detectors (Girshick et al., 2014; Ren et al., 2015; He et al., 2017) first generate region proposals, followed by classification and refinement. Pixel-based detectors (Redmon et al., 2016; Tian et al., 2025) directly classify and predict bounding boxes over pre-defined anchor boxes or pixels. Query-based detectors (Carion et al., 2020) adopt a transformer decoder to decode object queries to boxes. Despite the high detection accuracy, these close-set detectors are restricted to detecting the pre-defined categories and fail to generalize to unseen categories, limiting their applicability in real-world scenarios.

**Open-vocabulary object detection.** Open-vocabulary object detectors can recognize arbitrary categories by training with only the detection annotations of base categories. To achieve such generalization, existing methods typically introduce the language modality as the auxiliary, either by incorporating extra training data or by harnessing vision-language models. OVR-CNN (Zareian et al., 2021) pretrains the visual backbone on image-caption pairs and fine-tunes with detection annotations to learn OVOD. Some methods generate detection pseudo-labels to train the detector with a larger vocabulary, e.g., from image-level labels (Zhou et al., 2022), from image-caption data (Gao et al., 2022), from unlabeled images (Zhao et al., 2022). GLIP (Li et al., 2022) and GroundingDINO (Liu et al., 2024) regard the detection task as the grounding task and jointly learns object localization and vision-language alignment. Several works follow DETR-style detectors that detect objects via queries (Zang et al., 2022; Wu et al., 2023c; Li et al., 2023; Zhang et al., 2025; Wang et al., 2025a). Recently, utilizing pre-trained vision-language models for OVOD has become a popular approach, as they offer transferable image-text alignment for strong zero-shot recognition capability.

**Utilizing CLIP for OVOD.** CLIP (Radford et al., 2021), a powerful and widely adopted vision-language pre-training method, is capable of aligning a broad range of visual and textual concepts and demonstrates impressive zero-shot recognition performance. Various methods attempt to utilize CLIP for the OVOD task by transferring its knowledge to the detector. ViLD (Gu et al., 2021) distills CLIP's knowledge to the detector by aligning RoI features with CLIP features, and combining the scores of both CLIP and the detector via geometric mean. RegionCLIP (Zhong et al., 2022) adapts CLIP models for better region-level visual representations that enable fine-grained region-text alignment. EdaDet (Shi & Yang, 2023) preserves fine-grained local semantics by aligning dense image features to CLIP's semantic space. To avoid the need for costly retraining, F-VLM (Kuo et al., 2022) adopts a frozen pre-trained CLIP visual encoder as the backbone of the two-stage detector and trains only the detector head, while maintaining the score-level combination strategy. F-ViT (Wu et al., 2023b) further extends the framework by using the self-distilled ViT-based CLIP models as the backbone. DST-Det (Xu et al., 2023) treats background proposals as novel categories during training, and adopts a self-training strategy to enhance the detection performance of novel objects. VMCNet (Gao et al., 2025) adds a CNN branch besides the ViT-based CLIP backbone to leverage the strengths of different networks. Many recent works (Jing et al., 2024; Li et al., 2024; Qiu et al., 2025; Wang et al., 2025b; Xie et al., 2025) focus on bridging the distribution gap between image-level pretraining and region-level perception of ViT-based CLIP models, while still using the F-ViT framework that relies on score-level combination to enable open-vocabulary recognition. Unlike most works that focus on improving or utilizing CLIP for its image-text alignment ability to improve region-level classification, our MCT-Det utilizes CLIP's knowledge through a multi-level transfer strategy that extends the role of CLIP and contributes to both localization and classification. In this way, we effectively leverage the complementary strengths of CLIP and the detector to improve OVOD performance.

## 3 METHOD

In this section, we first briefly introduce the definition of the OVOD task and the motivation of our approach, followed by a detailed description of our proposed Multi-Level CLIP Transfer framework, which effectively transfers CLIP to the OVOD task through deep integrations at the feature, embedding, and score levels.

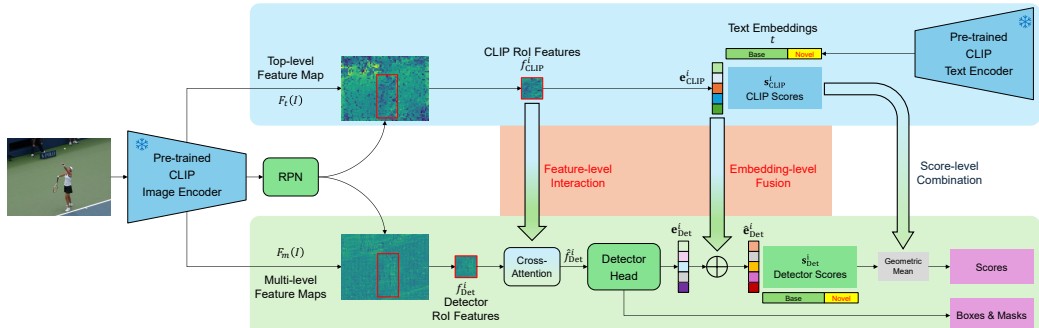

Figure 2: Overall architecture of MCT-Det.

## 3.1 PRELIMINARY

**Task formulation.** Given a set of base categories $C_B$ and a set of novel categories $C_N$, where $C_B \cap C_N = \emptyset$. During training, the open-vocabulary detector has access only to the detection annotations of $C_B$, while at test time it is required to detect arbitrary objects that belong to $C_B \cup C_N$ (Zareian et al., 2021). A common approach is to replace the classifier weights of conventional detectors with text embeddings of candidate category names, thereby performing open-vocabulary classification by computing similarity scores between the visual and textual embeddings.

**Motivation.** F-ViT (Wu et al., 2023b) extends Mask R-CNN (He et al., 2017) to the open-vocabulary setting by employing frozen CLIP vision encoders as the backbone and combining the CLIP and detector scores. However, there still exists a significant performance gap between novel and base categories in F-ViT. Based on the experimental analysis in Sec. 4.2, we conclude that the score-level combination serves as a trade-off between CLIP and detector scores, with only partial integration of CLIP's classification capability and the detector's localization capability. CLIP, through large-scale pretraining, possesses strong image-text alignment capabilities and excels at classification, but is weak in localization due to its global alignment pre-training task. In contrast, the detector, trained with annotations of base categories, is proficient in class-agnostic object localization but struggles to generalize its classification ability to novel categories. Thus, we are motivated to explore new transfer strategies that can better leverage the complementary strengths of CLIP and the detector.

## 3.2 OVERALL ARCHITECTURE

The overall architecture of MCT-Det is illustrated in Fig. 2. We build our framework upon F-ViT, retaining the frozen CLIP backbone and the score-level combination. On top of this baseline, we further introduce two additional transfer strategies: 1) the feature-level interaction that enriches the region features with global context for better localization, and 2) the embedding-level fusion that injects image-text alignment priors to region embeddings for better classification. Together with the score-level combination that trades off between classification and localization, the multi-level strategy effectively transfers CLIP's context, alignment, and generalization knowledge to the detector.

Given $I$ as the input image, the CLIP backbone first encodes the multi-scale feature maps $F_m(I)$ and the top-level feature map $F_t(I)$, then the RPN generates region proposals $b$ based on $F_m(I)$:

$$F_m(I) = \{F_1(I), F_2(i), ..., F_t(I)\} = \text{Backbone}(I)$$
$$b = \text{RPN}\Big(\phi\big(F_m(I)\big)\Big) \tag{1}$$

where $\phi(\cdot)$ stands for FPN. Subsequently, the region features of both detector and CLIP are cropped using RoIAlign. The ***feature-level interaction*** is introduced at this stage to transfer CLIP's global context knowledge, enriching region features with semantic information beyond the detector's original scope. Next, the RoI Head predicts class-agnostic boxes and masks, and encodes region embeddings based on integrated region features. The ***embedding-level fusion*** is applied at this stage to

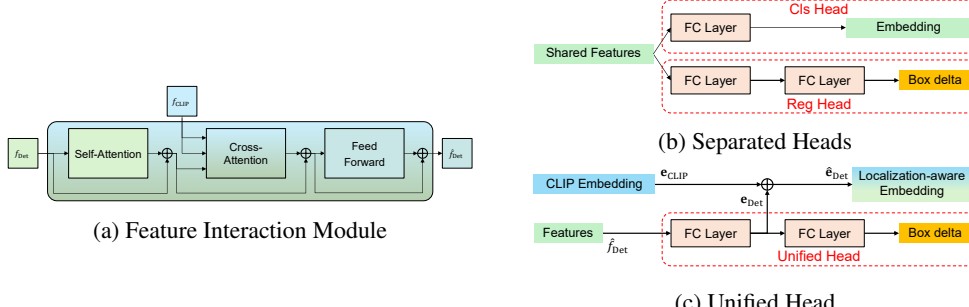

(a) Feature Interaction Module

(b) Separated Heads

(c) Unified Head

Figure 3: (a) The architecture of the feature interaction module consists of SA-CA-FFN. (b) The separated Classification and Regression Head, which share input features and generate embedding and box delta, respectively. (c) The proposed Unified Head, where the intermediate feature is repurposed as the localization-aware adaptation to CLIP embedding.

transfer CLIP's image-text alignment priors, yielding unified embeddings that inherit both CLIP's classification ability and the detector's localization ability. Finally, we retain the ***score-level combination*** in F-ViT to transfer CLIP's generalization capability, utilizing zero-shot classification results within the detection task.

## 3.3 Feature-Level Interaction

***Feature-Level Interaction*** *transfers CLIP's global context to the detector.* For each region proposal $b^i$, both CLIP's region features $f^i_{\text{CLIP}}$ and the detector's region features $f^i_{\text{Det}}$ are cropped using RoIAlign from different sources:

$$f^i_{\text{CLIP}} = \text{RoIAlign}\big(\psi\big(F_t(I)\big), b^i\big)$$
$$f^i_{\text{Det}} = \text{RoIAlign}\big(\phi\big(F_m(I)\big), b^i\big) \tag{2}$$

where $\psi(\cdot)$ denotes the projection layer of the CLIP visual encoder. $f^i_{\text{CLIP}}$ is cropped from the top-level feature map $F_t(I)$, which encodes rich global context and semantics but lacks sufficient fine-grained details crucial for detection tasks. On the contrary, $f^i_{\text{Det}}$ is cropped from multi-scale feature maps $F_m(I)$, enhancing localization-sensitive details at the expense of global information.

To enrich the detector features with global semantic context, we introduce a cross-attention module that enables interaction and fusion between CLIP and detector features. As illustrated in Fig. 3a, we adopt a decoder-style architecture (SA-CA-FFN), which consists of a self-attention block, a cross-attention block, and a feed-forward network, each equipped with residual connections to ensure stable feature transformation. Given the detector's region feature $f^i_{\text{Det}}$, we first capture intra-region dependencies with a self-attention operation. Subsequently, we employ CLIP's region feature $f^i_{\text{CLIP}}$ as the key and value in a cross-attention block to inject CLIP's global semantic information. The resulting fused feature is refined through a feed-forward network $\text{FFN}(\cdot)$ to obtain the final enhanced RoI feature $\tilde{f}^i_{\text{Det}}$. This process can be formulated as follows:

$$\hat{f}_{sa} = f^i_{\text{Det}} + \text{MHSA}(f^i_{\text{Det}})$$
$$\hat{f}_{ca} = \hat{f}_{sa} + \text{MHCA}(\hat{f}_{sa}, f^i_{\text{CLIP}}, f^i_{\text{CLIP}})$$
$$\tilde{f}^i_{\text{Det}} = \hat{f}_{ca} + \text{FFN}(\hat{f}_{ca}) \tag{3}$$

where $\text{MHSA}(\cdot)$ and $\text{MHCA}(\cdot)$ denote the multi-head self-attention block and multi-head cross-attention block, respectively. Through the cross-attention mechanism, global context information is dynamically integrated to enhance localization while preserving classification accuracy, resulting in a performance gain rather than a trade-off.

## 3.4 EMBEDDING-LEVEL FUSION

***Embedding-Level Fusion*** *transfers CLIP's image-text alignment priors to the detector.* Region embeddings refer to the final feature representations of candidate regions. In the F-ViT framework, both CLIP and the detector independently produce region embeddings. The detector's embeddings tend to overfit to base categories, while CLIP's embeddings fail to capture localization quality. As a result, they both fail to provide reliable detector scores.

We argue that an ideal region embedding should jointly encode the object category information and the proposal's localization quality, with both aspects can be reflected in the scores. To this end, we build a direct connection from CLIP's region embedding $\mathbf{e}_{\text{CLIP}}^i$ to the detector's region embedding $\mathbf{e}_{\text{Det}}^i$, yielding a unified embedding $\tilde{\mathbf{e}}_{\text{Det}}^i$ that achieves image-text alignment with localization-awareness. Moreover, as shown in Fig. 3c, instead of using separate branches (Classification Head and Regression Head in Fig. 3b), we design a unified head that utilizes the intermediate features as localization-aware adaptations of CLIP's embeddings, with the final output being box deltas:

$$\tilde{\mathbf{e}}_{\text{Det}}^i = \text{Norm}(\mathbf{e}_{\text{CLIP}}^i) + \text{Norm}(\mathbf{e}_{\text{Det}}^i) = \text{Norm}\big(\text{Mean}(f_{\text{CLIP}}^i)\big) + \text{Norm}\big(\text{RegFC}_1(\tilde{f}_{\text{Det}}^i)\big)$$
$$\Delta_{\text{box}} = \text{RegFC}_2\big(\text{ReLU}(\mathbf{e}_{\text{Det}}^i)\big) \tag{4}$$

where $\mathbf{e}_{\text{CLIP}}^i$ is obtained by applying mean pooling $\text{Mean}(\cdot)$ on $f_{\text{CLIP}}^i$, and $\mathbf{e}_{\text{Det}}^i$ denotes the intermediate feature obtained from the Regression Head, specifically the output of its first fully connected layer $\text{RegFC}_1(\cdot)$. $\text{Norm}(\cdot)$ represents $L_2$ normalization, as the two embeddings differ in scale and distribution, normalization aligns them and enables effective fusion. In this way, the detector studies class-agnostic localization-aware knowledge and refines CLIP embeddings according to the localization quality of proposals. By simplifying the detector's training objective, we alleviate overfitting and obtain reliable region embeddings. Moreover, the fused embeddings refine the scoring of candidate proposals and therefore benefit post-processing in the detection pipeline, which in turn improves detection accuracy.

## 3.5 SCORE-LEVEL COMBINATION

***Score-Level Combination*** *transfers CLIP's zero-shot classification capability to the detector.* Same as F-ViT, CLIP scores $\mathbf{s}_{\text{CLIP}}^i$ and detector scores $\mathbf{s}_{\text{Det}}^i$ are obtained through computing cosine similarity with text embeddings $\mathbf{t}_j$ of each class $j$:

$$\mathbf{s}_{\text{CLIP}}^i = \text{Softmax}\big(\frac{1}{T}[\cos(\mathbf{e}_{\text{CLIP}}^i, \mathbf{t}_{bg}), \cos(\mathbf{e}_{\text{CLIP}}^i, \mathbf{t}_1), \dots, \cos(\mathbf{e}_{\text{CLIP}}^i, \mathbf{t}_c)]\big)$$
$$\mathbf{s}_{\text{Det}}^i = \text{Softmax}\big(\frac{1}{\tau}[\cos(\tilde{\mathbf{e}}_{\text{Det}}^i, \mathbf{t}_{bg}), \cos(\tilde{\mathbf{e}}_{\text{Det}}^i, \mathbf{t}_1), \dots, \cos(\tilde{\mathbf{e}}_{\text{Det}}^i, \mathbf{t}_c)]\big) \tag{5}$$

where $\cos(a, b) = a^T b / (\|a\|\|b\|)$, and CLIP scores use a fixed temperature $T$ while detector scores adopt a learned temperature $\tau$. We feed the words of category names with predefined prompt templates (e.g., "a photo of the in the scene") into the CLIP text encoder, and average the embeddings over all templates to obtain text embeddings $\mathbf{t}$. We retain the score-level combination strategy in F-ViT, as the CLIP scores provide strong zero-shot classification capability that can classify a wide range of visual concepts. The final scores $\mathbf{s}^i$ are a combination of both the detector scores and the CLIP scores through geometric mean:

$$\mathbf{s}^i = \begin{cases} \mathbf{s}_{\text{CLIP}}^i(j)^{\alpha} \cdot \mathbf{s}_{\text{Det}}^i(j)^{(1-\alpha)} & \text{if } j \in C_B \\ \mathbf{s}_{\text{CLIP}}^i(j)^{\beta} \cdot \mathbf{s}_{\text{Det}}^i(j)^{(1-\beta)} & \text{if } j \in C_N \end{cases} \tag{6}$$

where $\alpha, \beta \in [0, 1]$ are the weights for base and novel categories, respectively. For the background, we use the text embedding of the word "background" and treat it as a base category. The combination coefficient depends only on whether a category $j$ belongs to the base set $C_B$ or novel set $C_N$ (i.e., whether it was seen during training). The final predicted label is obtained via an $\arg\max$ over the combined scores.

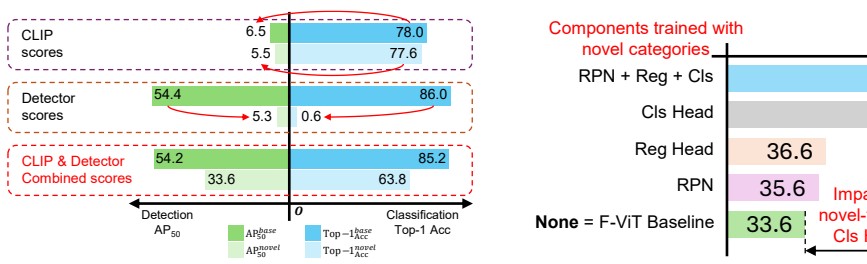

(a) Performance of CLIP and detector scores. (b) Impact of novel-category training.

Figure 4: Empirical motivation. (a) The respective detection and classification performance of CLIP and detector scores highlights their complementary strengths. (b) Training each component with novel-category annotations demonstrates the necessity of strengthening the classification head.

## 4 EXPERIMENTS

In this section, we present a comprehensive evaluation of the proposed method. We first describe the experimental setup, followed by an analysis of the score-level combination to justify our motivation. Next, we conduct ablation studies to verify the effectiveness of each proposed transfer strategy. Then, we compare our method with state-of-the-art approaches to demonstrate its competitiveness. Finally, we conduct cross-dataset transfer evaluation to further assess the generalization ability.

### 4.1 EXPERIMENT SETUP

**Benchmark details.** We evaluate our method on OV-COCO (Lin et al., 2014; Zareian et al., 2021) and OV-LVIS (Gupta et al., 2019; Gu et al., 2021) benchmarks.

- **OV-COCO benchmark.** OV-COCO divides COCO into 48 base categories and 17 novel categories. The model is trained on 48 base categories with 107,761 images and 665,387 instances, and then evaluated on all 65 categories with 4,836 images and 33,152 instances. We follow previous works to use the Average Precision (AP) of predicted boxes under the Intersection over Union (IoU) threshold $0.5$ of novel categories ($\mathrm{AP}_{50}^{novel}$) as the main metric.
- **OV-LVIS benchmark.** OV-LVIS treats 461 common and 405 frequent classes as base categories, and regards 337 rare classes as novel categories. The model is trained on 866 base categories with 100,170 images and 1,264,883 instances, then evaluated on all 1203 categories with 19,809 images and 244,707 instances. We follow previous works to use the mean AP of predicted masks across IoU thresholds from $0.5$ to $0.95$ of rare categories ($\mathrm{mAP}_r$) as the main metric.

**Implementation details.** We follow the settings of F-ViT (Wu et al., 2023b) for fair comparison. We adopt ViT-based EVA-CLIP models (Sun et al., 2023) fine-tuned with CLIPSelf (Wu et al., 2023b) as the backbone, and use Faster R-CNN (Ren et al., 2015) for OV-COCO and Mask R-CNN (He et al., 2017) for OV-LVIS, both are with FPN (Lin et al., 2017). Text embeddings are obtained by encoding each category name with multiple template prompts through the CLIP text encoder, and averaging the resulting vectors, including an additional embedding for the "background". Our experiments are conducted using 4 NVIDIA RTX 3090 GPUs with 8 samples per card, and the models are optimized using AdamW (Loshchilov & Hutter, 2017) with a learning rate of 5e-5 and weight decay of $0.1$. We train the model for 3 epochs on OV-COCO and 48 epochs on OV-LVIS. For feature-level interaction, we employ one SA-CA-FFN block with a head number of $4$. An exception is for training models with ViT-B on OV-COCO, where we use 16 samples per card and a learning rate of $1 \times 10^{-4}$. For score-level combination, we set $\alpha = 0.0$ and $\beta = 0.5$. More details are provided in Appendix A.1.

### 4.2 EMPIRICAL MOTIVATION

We first conduct experiments to validate our motivation. The training and testing are conducted on the OV-COCO and follow the settings of F-ViT, adopting ViT-B/16 (fine-tuned with CLIPSelf (Wu et al., 2023b) using image patches) as backbone and Faster R-CNN as detector.

Table 1: Ablation studies. We evaluate the effectiveness of each transfer strategy on detection performance, as well as the design of the detector head.

| Method | Transfer | | | Detector Heads | $AP_{50}^{novel}$ | $AP_{50}^{base}$ | $AP_{50}$ |
| | Feature | Embedding | Score | | | | |
| --- | --- | --- | --- | --- | --- | --- | --- |
| F-ViT | - | - | ✓ | | 33.6 | 54.2 | 48.8 |
| Variants | - | - | - | Reg & Cls | 5.3 | 54.4 | 41.6 |
| | ✓ | - | ✓ | | 35.2 | **55.9** | 50.5 |
| | - | ✓ | ✓ | | 41.3 | 54.4 | 51.0 |
| | ✓ | ✓ | ✓ | | 42.3 | 55.1 | 51.8 |
| MCT-Det | ✓ | ✓ | ✓ | Unified | **43.1** | 55.1 | **52.0** |

**Limitation of the score-level combination.** As formulated in Eq. 6, the final scores are combined as the geometric mean of CLIP scores and detector scores. To justify this design, we separately evaluate each score in two aspects: 1) the detection performance on novel categories ($AP_{50}^{novel}$), and 2) the Top-1 classification accuracy on ground-truth boxes of novel objects (Top-$1_{acc}^{novel}$). Results in Fig. 4a show a clear trade-off. CLIP scores exhibit strong classification ability across categories but are insensitive to the localization quality of proposals, resulting in poor detection accuracy when directly applied. In contrast, detector scores are reliable for both localization and classification of base categories, yet generalize poorly to novel categories. The score-level combination provides a reasonable solution to balance the two scores, but fails to fully exploit their complementary strengths.

**Impact of novel-category training on different components.** In the trainable part of F-ViT, the Region Proposal Network (RPN) and the Regression Head (Reg Head) generate and refine the proposals for localization, and the Classification Head (Cls Head) produces proposals' visual embeddings for classification. We train two F-ViT models with the same architecture but different data: 1) an open-vocabulary detector trained only on base categories, and 2) a conventional detector trained on both base and novel categories. By exchanging intermediate outputs, we can isolate the contribution of each component. As shown in Fig. 4b, the novel-category supervision significantly enhances the Cls Head and improves detection performance, while the benefit to localization-related components (RPN and Reg Head) is much less pronounced. This observation indicates that within the F-ViT framework, the localization-related components are robust to novel categories, while the classification head suffers from limited training data.

Our experiments illustrate the respective advantages of CLIP and the detector, as the former provides strong open-vocabulary recognition while the latter contributes robust localization. These observations inspire us to leverage their complementary strengths for better OVOD performance.

## 4.3 ABLATION STUDY

We conduct ablation studies on OV-COCO to verify the effectiveness of each proposed transfer strategy and the unified head. In both experiments, we adopt ViT-B/16 (fine-tuned with CLIPSelf using image patches) as the backbone, and follow the settings mentioned in Sec. 4.1.

To validate the effectiveness of each transfer strategy, we individually apply them on top of the F-ViT baseline, and the results are shown in Tab. 1. For the F-ViT baseline, the built-in score-level combination effectively utilizes CLIP's zero-shot classification capability in detection and improves $AP_{50}^{novel}$ from 5.3 to 33.6. By incorporating feature-level interaction, CLIP's global context information has effectively transferred to the detector, resulting in gains in overall precision on both novel ($+1.6$) and base ($+1.7$) categories. Introducing embedding-level fusion significantly improves $AP_{50}^{novel}$ to 42.3, indicating that our approach effectively utilized CLIP's image-text alignment capability for novel object classification in the detection task. Moreover, by integrating the proposed Multi-level CLIP Transfer strategy with the unified head design, MCT-Det achieves 43.1 in $AP_{50}^{novel}$, outperforming the baseline by 9.7. The experiments clearly demonstrate the effectiveness of the proposed multi-level transfer strategies and the benefit of coupling the classification and localization.

We additionally provide ablation experiments on the designs of the feature-level interaction module, the weights of controlling score-level combination, and the impact of using different CLIP backbones, which can be found in the Appendix A.2.

Table 2: Comparison with state-of-the-art OVOD methods.

(a) Results on OV-COCO benchmark.

| Method | Backbone | $AP_{50}^{novel}$ |
|---|---|---|
| OVR-CNN (Zareian et al., 2021) | RN50 | 22.8 |
| ViLD (Gu et al., 2021) | RN50 | 27.6 |
| F-VLM (Kuo et al., 2022) | RN50 | 28.0 |
| OV-DETR (Zang et al., 2022) | RN50 | 29.4 |
| VLDet (Lin et al., 2022) | RN50 | 32.0 |
| CFM-ViT (Kim et al., 2023) | ViT-L/16 | 34.1 |
| RegionCLIP (Zhong et al., 2022) | RN50x4 | 39.3 |
| BARON-Cap&KD (Wu et al., 2023a) | RN50x4 | 42.7 |
| CORA$^+$ (Wu et al., 2023c) | RN50x4 | 43.1 |
| OV-DQUO (Wang et al., 2025a) | RN50x4 | 45.6 |
| CCKT-Det++ (Zhang et al., 2025) | Swin-B | 46.0 |
| F-ViT + CLIPSelf (Wu et al., 2023b) | ViT-B/16 | 37.6 |
| MCT-Det + CLIPSelf (Ours) | ViT-B/16 | 44.4 |
| F-ViT + CLIPSelf (Wu et al., 2023b) | ViT-L/14 | 44.3 |
| DST-Det + CLIPSelf (Xu et al., 2023) | ViT-L/14 | 46.7 |
| F-ViT + R-SC-CLIPSelf (Qiu et al., 2025) | ViT-L/14 | 48.1 |
| F-ViT + VMCNet (Gao et al., 2025) | VMCNet-L | 48.5 |
| MCT-Det + CLIPSelf (Ours) | ViT-L/14 | **52.9** |

(b) Results on OV-LVIS benchmark.

| Method | Backbone | $mAP_r$ |
|---|---|---|
| ViLD (Gu et al., 2021) | RN50 | 16.6 |
| OV-DETR (Zang et al., 2022) | RN50 | 17.4 |
| RegionCLIP (Zhong et al., 2022) | RN50x4 | 22.0 |
| BARON (Wu et al., 2023a) | RN50 | 22.6 |
| VLDet (Lin et al., 2022) | Swin-B | 26.3 |
| CORA$^+$ (Wu et al., 2023c) | RN50x4 | 28.1 |
| F-VLM (Kuo et al., 2022) | RN50x64 | 32.8 |
| Detic (Zhou et al., 2022) | Swin-B | 33.8 |
| CFM-ViT (Kim et al., 2023) | ViT-L/16 | 33.9 |
| CoDet (Ma et al., 2023) | ViT-L/14 | 37.0 |
| OV-DQUO (Wang et al., 2025a) | ViT-L/14 | 39.3 |
| F-ViT + CLIPSelf (Wu et al., 2023b) | ViT-B/16 | 25.3 |
| MCT-Det + CLIPSelf (Ours) | ViT-B/16 | 27.6 |
| F-ViT + CLIPSelf (Wu et al., 2023b) | ViT-L/14 | 34.9 |
| DST-Det + CLIPSelf (Xu et al., 2023) | RN50x64 | 34.5 |
| F-ViT + R-SC-CLIPSelf (Qiu et al., 2025) | ViT-L/14 | 37.2 |
| F-ViT + VMCNet (Gao et al., 2025) | VMCNet-L | 38.4 |
| MCT-Det + CLIPSelf (Ours) | ViT-L/14 | **39.8** |

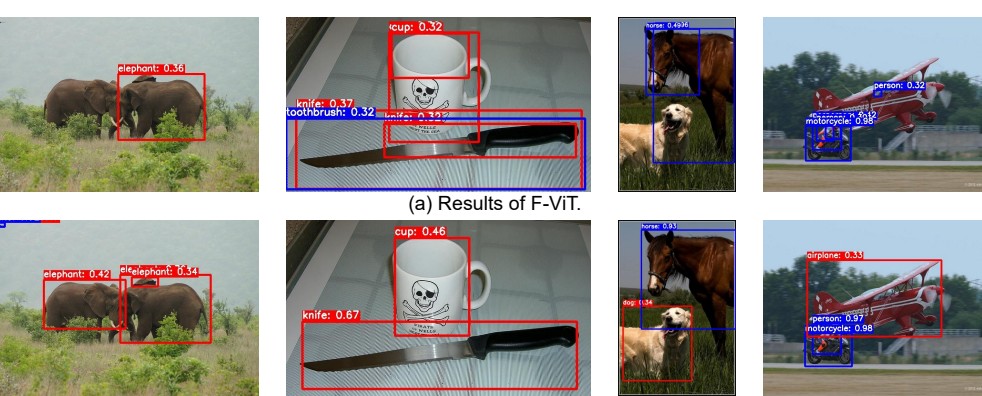

(a) Results of F-ViT.

(b) Results of MCT-Det.

Figure 5: Visualization of detection results on OV-COCO. We compare MCT-Det with the F-ViT baseline, both of which adopt ViT-B/16 as the backbone. Red boxes indicate novel categories, while blue boxes denote base categories.

## 4.4 BENCHMARK RESULTS

**Comparison on OV-COCO benchmark.** Tab. 2a shows the results of our approach with other state-of-the-art approaches on the OV-COCO benchmark. Using ViT-B/16 as the backbone, MCT-Det attains an $AP_{50}^{novel}$ of 44.5, surpassing the F-ViT by 6.9 and even outperforming the baseline with ViT-L/14. Scaling up the backbone further improves performance: with ViT-L/14, MCT-Det achieves 52.8 $AP_{50}^{novel}$ and outperforms F-ViT by 8.5, which significantly surpasses other methods.

**Comparison on OV-LVIS benchmark.** Tab. 2b shows the results of our approach with other state-of-the-art approaches on the OV-LVIS benchmark. Using ViT-B/16 as the backbone, our approach brings 1.9 $mAP_r$ gain compared to F-ViT. Using ViT-L/16, MCT-Det achieves state-of-the-art performance at 39.8 $mAP_r$ and outperformance the baseline by 4.9.

**Visualization.** In Fig. 5, we provide visualized detection results on OV-COCO. For clarity of demonstration, we filter the predicted bounding boxes using a score threshold of 0.3. Compared with the baseline, MCT-Det not only maintains strong detection performance on base objects but also exhibits notably higher precision in recognizing novel objects.

We also provide the complete benchmark results and more visualization results in Appendix A.3 and A.4, respectively.

Table 3: Cross-dataset evaluation of the model trained with OV-LVIS on COCO and Objects365.

| Method | COCO | | | Objects365 | | |
|---|---|---|---|---|---|---|
| | AP | AP50 | AP75 | AP | AP50 | AP75 |
| ViLD (Gu et al., 2021) | 36.6 | 55.6 | 39.8 | 11.8 | 18.2 | 12.6 |
| DetPro (Du et al., 2022) | 34.9 | 53.8 | 37.4 | 12.1 | 18.8 | 12.9 |
| BARON-KD (Wu et al., 2023a) | 36.2 | 55.7 | 39.1 | 13.6 | 21.0 | 14.5 |
| F-VLM (Kuo et al., 2022) | 39.8 | 61.6 | 43.8 | 17.7 | 27.4 | 19.1 |
| OV-DQUO (Wang et al., 2025a) | 39.2 | - | 42.5 | 18.4 | - | 19.6 |
| F-ViT + CLIPSelf (Wu et al., 2023b) | 40.5 | 63.8 | 44.3 | 19.5 | 31.3 | 20.7 |
| MCT-Det + CLIPSelf (Ours) | **40.6** | **64.3** | 43.5 | **21.4** | **33.9** | **22.9** |

## 4.5 CROSS-DATASET EVALUATION

To evaluate the generalization ability of MCT-Det, we conduct cross-dataset evaluation to assess whether the detector trained on one dataset can generalize to another dataset with disjoint categories and different data distributions. Specifically, we train MCT-Det (using ViT-L/14 as backbone) on OV-LVIS, and directly evaluate on the validation set of COCO (Lin et al., 2014) and Objects365v1 (Shao et al., 2019) by replacing the vocabulary without any further fine-tuning. Following F-VLM and F-ViT, we treat all categories as novel and use $\beta$ alone to combine the detector and CLIP scores. The vocabulary overlaps between OV-LVIS base categories and COCO/Objects365 are $91\%$ and $63\%$, respectively (Kuo et al., 2022). Hence, we set $\beta = 0.1$ on COCO and $\beta = 0.3$ on Objects365. The results are shown in Tab. 3.

On COCO, MCT-Det shows slight improvements over F-ViT in terms of AP and $AP_{50}$. This can be attributed to the high vocabulary overlap ($91\%$), where most categories were already encountered during training. In this case, the model relies more on the supervised-trained detector, rather than on the zero-shot CLIP. While on Objects365, MCT-Det achieves an AP of 21.4 and notably outperforms F-ViT across all metrics. The relatively low vocabulary overlap ($63\%$) poses greater challenges for the detector, yet our proposed multi-level transfer strategy effectively leverages CLIP's capabilities to support the recognition of unseen objects. These results demonstrate the robustness of our MCT-Det for open-vocabulary detection across diverse datasets.

## 5 CONCLUSION

In this paper, we propose a Multi-level CLIP Transfer framework for the OVOD task, namely **MCT-Det**. By incorporating *feature-*, *embedding-*, and *score-level* transfer strategies, our method effectively leverages the complementary strengths of CLIP's zero-shot classification and the detector's robust localization capabilities for the OVOD task. Extensive experiments demonstrate the effectiveness and scalability of MCT-Det, which significantly improves the detection precision on novel categories and outperforms the state-of-the-art methods.

## REPRODUCIBILITY STATEMENT

We provide the code in the supplementary material. We implement MCT-Det based on PyTorch and MMDetection (Chen et al., 2019), and the details are described in the paper (Sec. 4.1, Appendix A.1). All the base models (CLIP (Radford et al., 2021), EVA-CLIP (Sun et al., 2023), CLIPSelf and F-ViT (Wu et al., 2023b), Mask R-CNN (He et al., 2017)) and datasets (Lin et al., 2014; Gupta et al., 2019; Shao et al., 2019) used in this work are publicly available.

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

# A APPENDIX

## A.1 ADDITIONAL IMPLEMENTATION DETAILS

We build our detector upon F-ViT and follow the settings in (Wu et al., 2023b). On OV-COCO, we set the input image size to $640 \times 640$ for ViT-B/16 and $896 \times 896$ for ViT-L/14. On OV-LVIS, we set the input image size to $1024 \times 1024$ for ViT-B/16 and $896 \times 896$ for ViT-L/14. For data preprocessing, we use large-scale jittering of range $[0.1, 2.0]$ and random horizontal flip with a ratio of $0.5$ to augment input images during training, and apply zero-padding at the bottom and right of the images for non-square inputs during testing. For models employing *embedding-level fusion*, we set $\alpha = 0.0$ and $\beta = 0.5$. For other models, we follow F-ViT to set $\alpha = 0.1$ and $\beta = 0.8$. We initialize the learnable temperature $\tau$ of the detector with $50.0$, and set the fixed temperature $T$ of CLIP to $75.0$ on OV-COCO and $50.0$ on OV-LVIS. We apply a linear warm-up strategy during the first 250 iterations, and decay the learning rate at the target epoch by a factor of $0.1$. Specifically, on OV-COCO, the decay is applied after the 2nd epoch, while on OV-LVIS, it is applied at the 32nd and 40th epochs.

We adopt ViT-based EVA-CLIP models (Sun et al., 2023) fine-tuned with CLIPSelf (Wu et al., 2023b) as the backbone, which are self-distilled to align dense feature maps' regional features with the corresponding cropped images' global features. CLIPSelf provides two cropping strategies: 1) crop on fixed image patches (i.e., using image patches), and 2) crop on the proposals generated by class-agnostic RPN (i.e., using region proposals). For ablation experiments on OV-COCO and for benchmarking on OV-LVIS, we adopt the models fine-tuned using image patches as the backbone. As for benchmarking OV-COCO, we adopt the models fine-tuned using region proposals. All models are fine-tuned on their corresponding datasets.

## A.2 MORE ABLATION

We conduct more ablation studies to verify each proposed component. Unless specified, the experiments are performed on OV-COCO with ViT-B/16 CLIPSelf (finetuned using image patches) as the backbone.

**Ablation on feature-level interaction module.** We evaluate several designs for the feature-level interaction module that integrate $f_{\text{CLIP}}^i$ and $f_{\text{Det}}^i$, including the Add block, the Concat block, the iterative Attentional Feature Fusion (iAFF) block (Dai et al., 2021), the CA-FFN block, and our adopted SA-CA-FFN block. In the Add block, $f_{\text{CLIP}}^i$ is linearly transformed and then added to $f_{\text{Det}}^i$. In the Concat block, $f_{\text{CLIP}}^i$ and $f_{\text{Det}}^i$ are directly concatenated along the channel dimension. The iAFF block is built upon channel-wise attention and iteratively fuses $f_{\text{CLIP}}^i$ and $f_{\text{Det}}^i$:

$$\text{iAFF}(f_{\text{CLIP}}^i, f_{\text{Det}}^i) = \mathbf{M}(L(f_{\text{CLIP}}^i) + f_{\text{Det}}^i) \otimes L(f_{\text{CLIP}}^i) + (1 - \mathbf{M}(L(f_{\text{CLIP}}^i), f_{\text{Det}}^i)) \otimes f_{\text{Det}}^i \tag{7}$$

where $\mathbf{M}(\cdot)$ denotes the attentional weights generated by the multi-scale channel attention module, and $L(\cdot)$ denotes a linear transformation applied on $f_{\text{CLIP}}^i$ to align the channel dimensions. The CA-FFN block is based on our adopted SA-CA-FFN block by removing the self-attention layer. The SA-FFN block only computed self-attention of $f_{\text{Det}}^i$ and without integrating $f_{\text{CLIP}}^i$. We conduct evaluations of each module upon F-ViT with ViT-B/16 CLIPSelf (fine-tuned using image patches) as the backbone on OV-COCO, and additionally report their inference latency under fp32.

As shown in Tab. 4, the Add block improves the overall detection performance, suggesting that $f_{\text{CLIP}}^i$ provides beneficial global context to the detector. However, other interaction modules only enhance the performance of base categories, leaving novel categories nearly unchanged or even degraded. This limitation indicates the necessity of a more effective design to exploit this context without overfitting to base categories. The SA-CA-FFN block effectively addresses this issue, achieving notable improvements on both novel ($+1.6$) and base ($+1.7$) categories without introducing significant computation costs.

**Ablation on the unified head.** We measured the contribution of the unified head on detection performance and inference latency. As the results shown in Tab. 5, for both F-ViT and our MCT-Det, the

Table 4: Results of using different Feature Interaction Modules on OV-COCO.

| Backbone | Method | $AP_{50}^{novel}$ | $AP_{50}^{base}$ | $AP_{50}$ | Latency (ms) |
|---|---|---|---|---|---|
| F-ViT ViT-B/16 CLIPSelf using Image Patches | - | 33.6 | 54.2 | 48.8 | 60.1 |
| | Add | 34.4 (+0.8) | 54.7 (+0.5) | 49.4 (+0.6) | 62.4 |
| | Concat | 33.6 (+0.0) | 55.4 (+1.2) | 49.7 (+0.9) | 66.1 |
| | iAFF | 32.5 (-1.1) | 54.9 (+0.7) | 49.0 (+0.2) | 67.0 |
| | CA-FFN | 33.3 (-0.3) | 55.5 (+1.3) | 49.7 (+0.9) | 71.2 |
| | SA-FFN | 32.4 (-1.2) | 55.3 (+1.1) | 49.3 (+0.5) | 70.3 |
| | SA-CA-FFN | **35.2 (+1.6)** | **55.9 (+1.7)** | **50.5 (+1.7)** | 76.6 |

Table 5: Contribution of the unified detector head.

| Method | $AP_{50}^{novel}$ | Latency (ms) |
|---|---|---|
| F-ViT w/ Separate Heads | 33.6 | 60.1 |
| F-ViT w/ Unified Head | 34.4 | 60.0 |
| MCT-Det w/ Separate Heads | 42.3 | 76.6 |
| MCT-Det w/ Unified Head | 43.1 | 76.4 |

unified head exhibits a small reduction in inference latency, while consistently improving the detection precision (+0.8 $AP_{novel}^{50}$). Although this improvement is smaller than our two main contributions (i.e., feature- and embedding-level fusion), it still provides a measurable benefit and contributes to the overall performance.

**Ablation on score-level combination weights.** We further explore how the combination weights $\alpha$ and $\beta$ between CLIP and detector scores influences the detection performance. $\alpha$ affects base categories, and a larger $\alpha$ increases the contribution of detector scores. $\beta$ affects novel categories, and a larger $\beta$ increases the contribution of CLIP scores. We evaluate the different values of each weight individually by fixing the other at its optimal value (0.0 for $alpha$ and 0.5 for $beta$).

Based on the results in Tab. 6, the best detection performance is obtained with $\alpha = 0.0$ and $\beta = 0.5$, reflecting the fact that the detector is sufficient to produce reliable scores for base categories, whereas CLIP provides valuable support in identifying novel objects. We also observe that MCT-Det is relatively robust to $\beta$, as the embedding-level fusion has already injected the image-text alignment priors into the detector, and thus the detector scores themselves possess strong capability in classifying novel objects.

**Ablation on CLIP Backbones.** We evaluate the impact of using different CLIP backbones on MCT-Det. We employ various CLIP backbones with different scales, including CNN-based CLIP, ViT-based CLIP, and CLIP variants (i.e., SigLIP (Zhai et al., 2023), EVA-CLIP, CLIPSelf). As shown in Fig. 7, MCT-Det consistently yields significant improvements (with an average gain of about 8.0 on $AP_{50}^{novel}$) across diverse backbones, demonstrating the robustness and scalability of our method.

**Ablation on localization and classification.** To clearly demonstrate the improvements of proposed feature-level interaction and embedding-level fusion, we separately evaluate localization and classification metrics, and the results are listed in Tab. 8.

- **Localization:** We adopt a binary target/background metric to measure the localization ability (we denote them as Loc-AP/AR). The results show clear improvements for both the feature-level interaction and embedding-level fusion, indicating they both contribute to a better localization performance.
- **Classification:** We evaluate the detector's classification performance by using only detector scores on novel categories and measuring: (1) Top-1 classification accuracy on the ground-truth boxes, and (2) the detection precision. As the results show, the feature-level interaction brings little improvement for novel-category classification, whereas embedding-level fusion leads to a significant gain.

Table 6: Results of using different weights for score-level combination on OV-COCO.

(a) Combination weights of novel categories. We fix $\alpha = 0.0$.

| $\beta$ | $AP_{50}^{novel}$ | $AP_{50}^{base}$ | $AP_{50}$ |
|---|---|---|---|
| 0.0 | 38.7 | 55.1 | 50.8 |
| 0.1 | 40.2 | 55.1 | 51.2 |
| 0.2 | 41.5 | 55.1 | 51.5 |
| 0.4 | 42.8 | 55.1 | 51.9 |
| 0.5 | **43.1** | **55.1** | **52.0** |
| 0.6 | 43.1 | 55.0 | 51.9 |
| 0.8 | 41.3 | 54.9 | 51.4 |
| 0.9 | 38.8 | 54.9 | 50.7 |
| 1.0 | 5.9 | 54.5 | 41.8 |

(b) Combination weights of base categories. We fix $\beta = 0.5$.

| $\alpha$ | $AP_{50}^{novel}$ | $AP_{50}^{base}$ | $AP_{50}$ |
|---|---|---|---|
| 0.0 | **43.1** | **55.1** | **52.0** |
| 0.1 | 43.1 | 54.8 | 51.7 |
| 0.2 | 43.0 | 54.4 | 51.4 |
| 0.4 | 43.0 | 53.1 | 70.5 |
| 0.5 | 43.0 | 52.2 | 49.8 |
| 0.6 | 42.9 | 51.1 | 49.0 |
| 0.8 | 42.0 | 47.3 | 45.9 |
| 0.9 | 40.7 | 43.2 | 42.6 |
| 1.0 | 37.5 | 6.7 | 14.8 |

Table 7: Results of MCT-Det using different backbones on OV-COCO.

| Method | Backbone | $AP_{50}^{novel}$ | $AP_{50}^{base}$ | $AP_{50}$ |
|---|---|---|---|---|
| F-ViT | OpenAI-CLIP | 16.0 | 36.9 | 31.4 |
| MCT-Det | ViT-B/16 | 23.5 (+7.5) | 41.0 | 36.4 |
| F-ViT | SigCLIP | 16.3 | 40.6 | 34.2 |
| MCT-Det | ViT-B/16 | 23.0 (+6.7) | 42.2 | 37.2 |
| F-ViT | EVA-CLIP | 17.5 | 41.0 | 34.9 |
| MCT-Det | ViT-B/16 | 26.9 (+9.4) | 43.8 | 39.4 |
| F-ViT | EVA-CLIPSelf (Image Patch) | 33.6 | 54.2 | 48.8 |
| MCT-Det | ViT-B/16 | 43.1 (+9.7) | 55.1 | 52.0 |
| F-ViT | EVA-CLIPSelf (Region Proposal) | 37.6 | 54.9 | 50.4 |
| MCT-Det | ViT-B/16 | 44.4 (+6.8) | 57.0 | 53.7 |
| F-ViT | EVA-CLIPSelf (Image Patch) | 38.4 | 60.6 | 54.8 |
| MCT-Det | ViT-L/14 | 50.5 (+12.1) | 63.2 | 59.9 |
| F-ViT | EVA-CLIPSelf (Region Proposal) | 44.3 | 64.1 | 59.0 |
| MCT-Det | ViT-L/14 | 52.9 (+8.6) | 65.7 | 62.4 |
| F-ViT | OpenAI-CLIP | 24.1 | 44.9 | 39.5 |
| MCT-Det | RN50 | 30.5 (+6.4) | 46.2 | 42.1 |
| F-ViT | OpenAI-CLIP | 29.4 | 49.9 | 44.6 |
| MCT-Det | RN50x4 | 34.6 (+5.2) | 50.4 | 46.3 |

==These experiments indicate that the feature-level interaction mainly enhances localization capability, while the embedding-level fusion strengthens classification and refines proposal scoring, which in turn feeds back to further improve localization performance.==

**The class-agnostic localization capability of the detector.** ==To quantitatively assess whether the detector can localize novel categories when only trained on base categories, we evaluated the average recall on novel categories of our MCT-Det. We also directly measure the detector's localization ability via the mentioned Loc-AP/AR metrics. As the results show in Tab. 9, compared to the novel-trained detector, the base-trained detector still maintains a high recall on novel objects (72.1 vs 80.1, about 10% drop), while its AP suffers a much larger decline (43.1 vs 62.7, about 30% drop). Meanwhile, the improvements in Loc-AP (32.0 vs 33.6) and Loc-AR (47.1 vs 48.4) metrics are relatively small, indicating that the detector already acquires sufficiently class-agnostic localization ability from base-category training. These observations support that the detector provides reliable class-agnostic localization ability.==

Table 8: Effects on localization and classification.

| Method | Loc-AP | Loc-AR | Using Only Detector Scores | |
| --- | --- | --- | --- | --- |
| | | | Top-1$^{novel}_{acc}$ | AP$^{novel}_{50}$ |
| F-ViT | 25.6 | 44.6 | 0.6 | 5.3 |
| F-ViT w/ Feat-level | 29.8 | 45.9 | 2.2 | 5.0 |
| F-ViT w/ Embed-level | 30.6 | 46.2 | 57.2 | 38.5 |
| MCT-Det (ours) | 32.0 | 47.1 | 58.3 | 38.7 |

Table 9: The class-agnostic localization capability of the detector.

| Training Data | AP$^{novel}_{50}$ | AR$^{novel}_{50}$@100 | Loc-AP | Loc-AR |
| --- | --- | --- | --- | --- |
| Base | 43.1 | 72.1 | 32.0 | 47.1 |
| Base+Novel | 62.7 | 80.1 | 33.6 | 48.4 |

## A.3 COMPLETE BENCHMARK RESULTS

We present the complete benchmark results of MCT-Det, including detection performance on both novel and base categories. Results on OV-COCO are shown in Tab. 10, where we additionally report AP$^{base}_{50}$ and overall AP$_{50}$ of boxes. Results of OV-LVIS are shown in Tab. 11, where we additionally report mAP$_c$, mAP$_f$, and overall mAP or masks. MCT-Det not only achieves state-of-the-art performance on novel categories, but also maintains competitive precision on base categories. The results demonstrate the effectiveness of the proposed transfer strategies and validate the overall robustness and generalization capability of MCT-Det.

## A.4 MORE VISUALIZATION

We provide more visualization results of MCT-Det on OV-LVIS using ViT-B/16 (fine-tuned with CLIPSelf using image patches) as the backbone. As illustrated in Fig. 6, our MCT-Det produces accurate bounding boxes and segmentation masks across both base and novel objects, demonstrating its robust generalization and effectiveness in the OVOD task.

## A.5 USAGE OF LARGE LANGUAGE MODELS

We use Large Language Models (LLMs) to help improve readability, polish writing, and refine grammar to enhance the overall presentation, while preserving the original meaning. **Importantly, LLMs are *not* involved in the design of methods, experiments, or analysis.** All scientific insights, including core ideas, module designs, experiments, analysis, and conclusions, remain entirely the work of the authors. We have ensured that the text generated with the assistance of LLMs adheres to the ICLR Code of Ethics and does not involve plagiarism or scientific misconduct.

Table 10: Complete results of OV-COCO.

| Method | Backbone | $AP50^{novel}$ | $AP50^{base}$ | $AP_{50}$ |
|---|---|---|---|---|
| OVR-CNN (Zareian et al., 2021) | RN50 | 22.8 | 46.0 | 39.9 |
| ViLD (Gu et al., 2021) | RN50 | 27.6 | 59.5 | 51.2 |
| F-VLM (Kuo et al., 2022) | RN50 | 28.0 | 43.7 | 39.6 |
| OV-DETR (Zang et al., 2022) | RN50 | 29.4 | 61.0 | 52.7 |
| VLDet (Lin et al., 2022) | RN50 | 32.0 | 50.6 | 45.8 |
| DK-DETR (Li et al., 2023) | RN50 | 32.3 | - | 61.1 |
| CFM-ViT (Kim et al., 2023) | ViT-L/16 | 34.1 | - | 46.0 |
| OADP (Wang et al., 2023) | RN50 | 35.6 | 55.8 | 50.5 |
| RegionCLIP (Zhong et al., 2022) | RN50x4 | 39.3 | 61.6 | 55.7 |
| LP-OVOD (Pham et al., 2024) | RN50 | 40.5 | 60.5 | 55.2 |
| DITO (Kim et al., 2024) | ViT-L/16 | 40.8 | - | 50.3 |
| BARON-Cap&KD (Wu et al., 2023a) | RN50x4 | 42.7 | 54.9 | 51.7 |
| CORA$^+$ (Wu et al., 2023c) | RN50x4 | 43.1 | 60.9 | 56.2 |
| OV-DQUO (Wang et al., 2025a) | RN50x4 | 45.6 | - | - |
| CCKT-Det++ (Zhang et al., 2025) | Swin-B | 46.0 | - | 46.2 |
| F-ViT + CLIPSelf (Wu et al., 2023b) | ViT-B/16 | 37.6 | 54.9 | 50.4 |
| MCT-Det + CLIPSelf (Ours) | ViT-B/16 | 44.4 | 57.0 | 53.7 |
| F-ViT + CLIPSelf (Wu et al., 2023b) | ViT-L/14 | 44.3 | 64.1 | 59.0 |
| F-ViT + DeCLIP (Wang et al., 2025b) | ViT-L/14 | 46.2 | - | - |
| DST-Det + CLIPSelf (Xu et al., 2023) | ViT-L/14 | 46.7 | 61.9 | 58.0 |
| F-ViT + R-SC-CLIPSelf (Qiu et al., 2025) | ViT-L/14 | 48.1 | 65.4 | 60.8 |
| F-ViT + VMCNet (Gao et al., 2025) | VMCNet-L | 48.5 | - | - |
| MCT-Det + CLIPSelf (Ours) | ViT-L/14 | **52.9** | **65.7** | **62.4** |

Table 11: Complete results of OV-LVIS.

| Method | Backbone | mAP$r$ | mAP$c$ | mAP$_f$ | mAP |
|---|---|---|---|---|---|
| ViLD (Gu et al., 2021) | RN50 | 16.6 | 24.6 | 30.3 | 25.5 |
| OV-DETR (Zang et al., 2022) | RN50 | 17.4 | 25.0 | 32.5 | 26.6 |
| LP-OVOD (Pham et al., 2024) | RN50 | 19.3 | 26.1 | 29.4 | 26.2 |
| DK-DETR (Li et al., 2023) | RN50 | 20.5 | 28.9 | 35.4 | 30.0 |
| OADP (Wang et al., 2023) | RN50 | 21.7 | 26.3 | 29.0 | 26.6 |
| RegionCLIP (Zhong et al., 2022) | RN50x4 | 22.0 | 32.1 | 36.9 | 32.3 |
| BARON (Wu et al., 2023a) | RN50 | 22.6 | - | - | - |
| VLDet (Lin et al., 2022) | Swin-B | 26.3 | 39.4 | 41.9 | 38.1 |
| CORA$^+$ (Wu et al., 2023c) | RN50x4 | 28.1 | - | - | - |
| F-VLM (Kuo et al., 2022) | RN50x64 | 32.8 | - | - | 34.9 |
| Detic (Zhou et al., 2022) | Swin-B | 33.8 | - | - | 40.7 |
| CFM-ViT (Kim et al., 2023) | ViT-L/16 | 33.9 | - | - | 36.6 |
| DST-Det (Xu et al., 2023) | RN50x64 | 34.5 | 33.7 | 33.5 | 34.6 |
| CoDet (Ma et al., 2023) | ViT-L/14 | 37.0 | 46.3 | 46.3 | 44.7 |
| OV-DQUO (Wang et al., 2025a) | ViT-L/14 | 39.3 | - | - | - |
| F-ViT + CLIPSelf (Wu et al., 2023b) | ViT-B/16 | 25.3 | 21.8 | 29.1 | 25.2 |
| MCT-Det + CLIPSelf (Ours) | ViT-B/16 | 27.6 | 26.6 | 29.1 | 27.8 |
| F-ViT + CLIPSelf (Wu et al., 2023b) | ViT-L/14 | 34.9 | 34.6 | 35.6 | 35.1 |
| F-ViT + DeCLIP (Wang et al., 2025b) | ViT-L/14 | 37.2 | - | - | - |
| F-ViT + R-SC-CLIPSelf (Qiu et al., 2025) | ViT-L/14 | 37.2 | 37.2 | 37.1 | 37.2 |
| F-ViT + VMCNet (Gao et al., 2025) | VMCNet-L | 38.4 | - | - | - |
| MCT-Det + CLIPSelf (Ours) | ViT-L/14 | **39.8** | 35.7 | 35.0 | 36.2 |

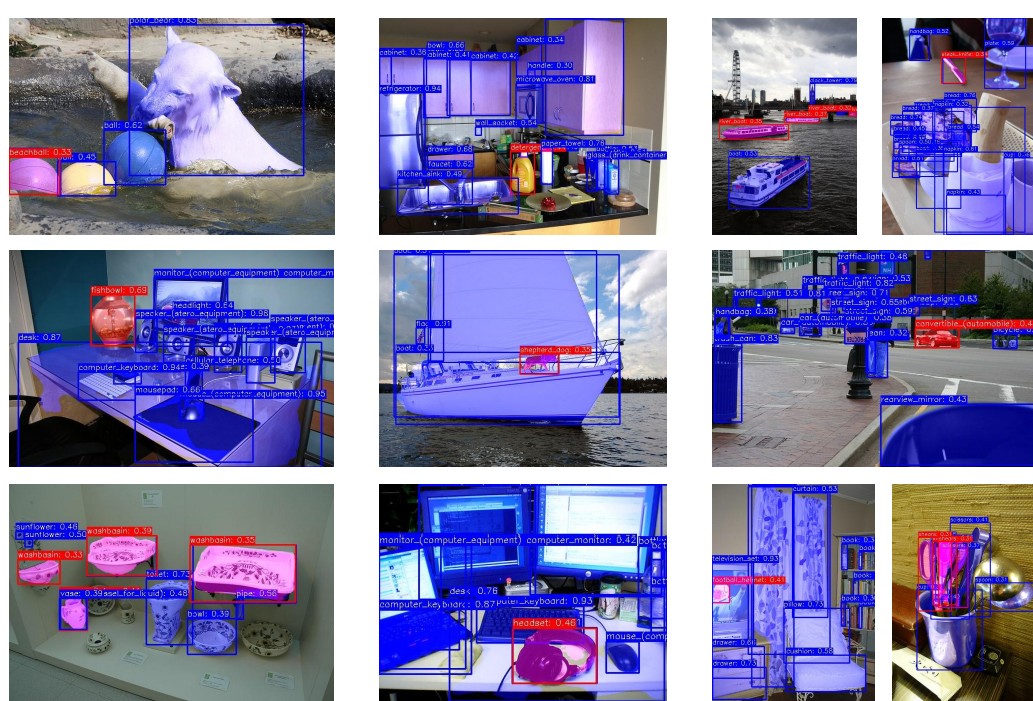

Figure 6: Visualization of MCT-Det's detection results on OV-LVIS. Red boxes and masks indicate novel categories, and blue boxes and masks represent base categories.

