# OpenReview forum: "Multi-Level CLIP Transfer for Open-Vocabulary Object Detection"
_ICLR.cc/2026/Conference — Submitted to ICLR 2026_

### Official Review · Reviewer_bUuB · 2025-10-30

**Soundness:** 3
**Presentation:** 4
**Contribution:** 3
**Rating:** 6
**Confidence:** 4

**Summary:**

This paper proposes a multi-level (feature level, embedding level, score level) knowledge transfer framework, which leverages CLIP to improve the localization and classification ability of the detector for novel classes.

**Strengths:**

**Originality**:
- The motivation for this paper is very clear, and Figure 1 intuitively illustrates the current problems in open-vocabulary object detection.

**Clarity**:
- This paper clearly elaborates the problems with existing open-vocabulary object detection methods, as well as the solutions it proposes to address them.

**Significance**:
- This paper addresses a key problem in open-vocabulary object detection and makes a significant contribution to this field.

**Weaknesses:**

- In Table 2, it seems that the proposed MCT-Det can only be attached to CLIPSelf, which may limit the applicability of MCT-Det.
- In Table 2, the proposed method is effectively only compared with F-ViT and DST-Det. The other methods use different backbones, resulting in an unfair comparison. I hope the authors can clarify this: are there no other methods available for a fair comparison?
- I think the title "CLIP Transfer" is not appropriate. The paper isn't about transferring CLIP itself, but about transferring knowledge from CLIP.

**Questions:**

- This paper uses AP50 and AP75 as evaluation metrics. However, considering the experiments aim to verify that "CLIP can improve the detector's localization ability," is there any metric that directly reflects localization capability? For example, by calculating AP using only binary target/background classification?

---

> ### Author Response · Authors · 2025-12-02
> **Response to Reviewer bUuB 1/2**
>
> We thank Reviewer bUuB for the effort in reviewing our work and providing supportive comments. We also appreciate the reviewer’s acknowledgment of the **originality**, **clarity**, and **significance** of our work. We address each concern or question below.
>
> ##
> > **W1:** In Table 2, it seems that the proposed MCT-Det can only be attached to CLIPSelf, which may limit the applicability of MCT-Det.
>
> We thank the reviewer for pointing out this concern.
>
> **MCT-Det is *not* tied to the specific training paradigm or the architecture of the CLIP backbone.** It is compatible with both original CLIP and CLIP variants. To address the reviewer's concern, we evaluate MCT-Det with 5 different CLIP backbones on OV-COCO. As the results show, MCT-Det achieves consistent improvements across various backbones, demonstrating its effectiveness and broad applicability.
>
> We include these results in Appendix A.2.
>
> | Method            | Backbone                 | AP50_novel    | AP50_base | AP50 |
> |-------------------|---------------------------|---------------|:-:|:-:|
> | F-ViT             | OpenAI-CLIP ViT-B/16      | 16.0          | 36.9      | 31.4 |
> | **MCT-Det (ours)**| OpenAI-CLIP ViT-B/16      | 23.5 (+7.5)   | 41.0      | 36.4 |
> | F-ViT             | SigLIP ViT-B/16           | 16.3          | 40.6      | 34.2 |
> | **MCT-Det (ours)**| SigLIP ViT-B/16           | 23.0 (+6.7)   | 42.2      | 37.2 |
> | F-ViT             | EVA-CLIP ViT-B/16         | 17.5          | 41.0      | 34.9 |
> | **MCT-Det (ours)**| EVA-CLIP ViT-B/16         | 26.9 (+9.4)   | 43.8      | 39.4 |
> | F-ViT             | OpenAI-CLIP RN50          | 24.1          | 44.9      | 39.5 |
> | **MCT-Det (ours)**| OpenAI-CLIP RN50          | 30.5 (+6.4)   | 46.2      | 42.1 |
> | F-ViT             | OpenAI-CLIP RN50x4        | 29.4          | 49.9      | 44.6 |
> | **MCT-Det (ours)**| OpenAI-CLIP RN50x4        | 34.6 (+5.2)   | 50.4      | 46.3 |
>
>
>
> ##
>
> > **W2:** In Table 2, the proposed method is effectively only compared with F-ViT and DST-Det. The other methods use different backbones, resulting in an unfair comparison. I hope the authors can clarify this: are there no other methods available for a fair comparison?
>
> We appreciate the reviewer for raising this fairness concern.
>
> We agree that a fair comparison to our method requires **using the same backbone** and **only modifying the detection framework**. However, different OVOD methods often adopt different backbones, detection frameworks, training strategies, or datasets, which makes fully fair comparisons across methods difficult to achieve. In OVOD research [1-5], it is common to provide fully apple-to-apple comparisons with a small number of closely related baselines, while also reporting direct comparisons with other methods.
>
> Since MCT-Det is built on top of F-ViT, applying our modifications to other frameworks would require many adaptations and efforts, which cannot be completed within the rebuttal period. Therefore, we follow the community’s common practice by comparing our MCT-Det to:
>
> * (1) F-ViT [4] and DST-Det [5], which, to the best of our knowledge, are the only existing methods that satisfy the criteria for a fully fair comparison.
> * (2) Other methods that use different backbones or detection frameworks, we include these comparisons to demonstrate the overall performance of our approach on the OVOD task.
>
>
> > **W3:** I think the title "CLIP Transfer" is not appropriate. The paper isn't about transferring CLIP itself, but about transferring knowledge from CLIP.
>
> We thank the reviewer for this suggestion.
>
> We agree that our method focuses on transferring knowledge from CLIP to the detector, rather than transferring CLIP itself.
> We revise the title from ***Multi-Level CLIP Transfer for Open-Vocabulary Object Detection*** to ***Multi-Level CLIP Knowledge Transfer for Open-Vocabulary Object Detection*** to more precisely reflect this intention.

---

> > ### Author Response · Authors · 2025-12-02
> > **Response to Reviewer bUuB 2/2**
> >
> > > **Q1:** This paper uses AP50 and AP75 as evaluation metrics. However, considering the experiments aim to verify that "CLIP can improve the detector's localization ability," is there any metric that directly reflects localization capability? For example, by calculating AP using only binary target/background classification?
> >
> > We thank the reviewer for this insightful suggestion.
> >
> > The suggested binary target/background metric is indeed a meaningful way to directly evaluate localization quality.
> > We therefore evaluate this metric on OV-COCO and measure both the average precision (we denote it as Loc-AP) and average recall (Loc-AR) across IoU thresholds from 0.5 to 0.95.
> >
> > The results show that both the feature-level interaction and the embedding-level fusion lead to consistent improvements in AP and AR.
> > These findings further confirm that CLIP features and embeddings can enhance the detector’s localization capability.
> >
> > We include the results and corresponding analysis in Appendix A.2.
> >
> > | Method              | Loc-AP | Loc-AR |
> > |---------------------|:-:|:-:|
> > | F-ViT               | 25.6   | 44.6   |
> > | F-ViT w/ Feat-level | 29.8   | 45.9   |
> > | F-ViT w/ Embed-level| 30.6   | 46.2   |
> > | **MCT-Det (ours)**  | 32.0   | 47.1   |
> >
> >
> > **References**
> >
> > [1] Wu, S., Zhang, W., Jin, S., Liu, W. and Loy, C.C., 2023. Aligning bag of regions for open-vocabulary object detection. In Proceedings of the IEEE/CVF conference on computer vision and pattern recognition (pp. 15254-15264).
> >
> > [2] Wu, X., Zhu, F., Zhao, R. and Li, H., 2023. Cora: Adapting clip for open-vocabulary detection with region prompting and anchor pre-matching. In Proceedings of the IEEE/CVF conference on computer vision and pattern recognition (pp. 7031-7040).
> >
> > [3] Wang, J., Chen, B., Kang, B., Li, Y., Xian, W., Chen, Y. and Xu, Y., 2025, April. Ov-dquo: Open-vocabulary detr with denoising text query training and open-world unknown objects supervision. In Proceedings of the AAAI Conference on Artificial Intelligence (Vol. 39, No. 7, pp. 7762-7770).
> >
> > [4] Wu, S., Zhang, W., Xu, L., Jin, S., Li, X., Liu, W. and Loy, C.C., 2023. Clipself: Vision transformer distills itself for open-vocabulary dense prediction. arXiv preprint arXiv:2310.01403.
> >
> > [5] Xu, S., Li, X., Wu, S., Zhang, W., Tong, Y. and Loy, C.C., 2023. Dst-det: Simple dynamic self-training for open-vocabulary object detection. arXiv preprint arXiv:2310.01393.

---

### Official Review · Reviewer_aQPQ · 2025-10-31

**Soundness:** 2
**Presentation:** 2
**Contribution:** 2
**Rating:** 4
**Confidence:** 4

**Summary:**

The paper finds that CLIP encodes global semantics for accurate classification but exhibits limited sensitivity to localization precision when scoring proposals, whereas the detector provides robust localization yet tends to misclassify novel objects as background. To leverage the complementary strengths of CLIP and the detector, the paper transfers the knowledge from CLIP to the detector from feature-level interaction, embedding-level fusion, and score-level combination. The resulting model outperforms its baseline F-ViT on OV-COCO and OV-LVIS.

**Strengths:**

1. The paper clearly demonstrates its motivation with a separate section with detailed experiment analysis, demonstrating CLIP is good at classification while the detector is good at localization, and training on the cls head with novel classes is the key to achieving high novel class performance.
2. The method is implemented on various backbones, showing that the method is scalable with the size of the backbone.
3. The ablation studies show that the performance is relatively robust to the combination weights on both novel classes and base classes, demonstrating that the method is not sensitive to the hyperparameters.

**Weaknesses:**

1. The contribution is incremental, as distilling or combining CLIP features for classification is well explored, including but not limited to  [1,2,3,4]. All of these papers also aim to combine the classification ability from CLIP and the localization ability from the detector. The paper does not compare with them to highlight the unique advantages of the proposed method.
2. The CLIP features in feature-level interaction and embedding-level fusion are the same (the only difference is the mean operation). But the benefits differ: the former is for better localization while the latter is for better classification. I don't get why the same feature can have totally different advantages. And there are no experimental results or analysis to support the claim.
3. Further, it is a consensus that CLIP is pretrained with an image-level objective thus it is not sensitive to the quality of bounding boxes [5,6,7]. The claim that feature-level interaction can improve the localization ability does not make sense.

[1] Open-vocabulary Object Detection via Vision and Language Knowledge Distillation. In ICLR 2022.

[2] Object-Aware Distillation Pyramid for Open-Vocabulary Object Detection. In CVPR 2023.

[3] EdaDet: Open-Vocabulary Object Detection Using Early Dense Alignment. In ICCV 2023.

[4] A Hierarchical Semantic Distillation Framework for Open-Vocabulary Object Detection. In TMM 2025.

[5] RegionCLIP: Region-based Language-Image Pretraining. In CVPR 2022.

[6] Frozen-DETR: Enhancing DETR with Image Understanding from Frozen Foundation Models. In NeurIPS 2024.

[7] CLIPSelf: Vision Transformer Distills Itself for Open-Vocabulary Dense Prediction. In ICLR 2024.

**Questions:**

Please see the weaknesses above.

---

> ### Author Response · Authors · 2025-12-02
> **Response to Reviewer aQPQ 1/2**
>
> We thank Reviewer aQPQ for the effort in reviewing our work and providing constructive feedback. We appreciate the reviewer's recognition of our **motivation analysis** and **the scalability and robustness** demonstrated in our experiments. We address each concern or question below.
>
>
> ##
> > **W1:** The contribution is incremental, as distilling or combining CLIP features for classification is well explored, including but not limited to [1,2,3,4]. All of these papers also aim to combine the classification ability from CLIP and the localization ability from the detector. The paper does not compare with them to highlight the unique advantages of the proposed method.
> >
> > [1] Open-vocabulary Object Detection via Vision and Language Knowledge Distillation. In ICLR 2022.
> >
> > [2] Object-Aware Distillation Pyramid for Open-Vocabulary Object Detection. In CVPR 2023.
> >
> > [3] EdaDet: Open-Vocabulary Object Detection Using Early Dense Alignment. In ICCV 2023.
> >
> > [4] A Hierarchical Semantic Distillation Framework for Open-Vocabulary Object Detection. In TMM 2025.
>
> We thank the reviewer for pointing out this concern.
>
> The key difference from previous works is that our transfer strategy not only leverages CLIP for region-level classification, but also transfers CLIP’s knowledge into the detector’s internal representations, allowing it to benefit both localization and classification.
>
> Prior works primarily distill or utilize CLIP for its image-text alignment ability to improve region-level classification. In contrast, our multi-level transfer strategy extends the role of CLIP to enhance the detector’s representations to improve the stability and discriminability of RoI features and embeddings, which in turn contribute to both localization and classification.
> Our experiments also prove the value of transferring generalized cross-modal representations into a specialized recognition task.
>
> We include the comparisons of MCT-Det to these prior works in Section 2.

---

> > ### Author Response · Authors · 2025-12-02
> > **Response to Reviewer aQPQ 2/2**
> >
> > > **W2:** The CLIP features in feature-level interaction and embedding-level fusion are the same (the only difference is the mean operation). But the benefits differ: the former is for better localization while the latter is for better classification. I don't get why the same feature can have totally different advantages. And there are no experimental results or analysis to support the claim.
> > >
> > > **W3:** Further, it is a consensus that CLIP is pretrained with an image-level objective thus it is not sensitive to the quality of bounding boxes [5,6,7]. The claim that feature-level interaction can improve the localization ability does not make sense.
> > >
> > > [5] RegionCLIP: Region-based Language-Image Pretraining. In CVPR 2022.
> > >
> > > [6] Frozen-DETR: Enhancing DETR with Image Understanding from Frozen Foundation Models. In NeurIPS 2024.
> > >
> > > [7] CLIPSelf: Vision Transformer Distills Itself for Open-Vocabulary Dense Prediction. In ICLR 2024.
> >
> > We thank the reviewer for raising this question.
> >
> > Although the CLIP features used in feature-level and embedding-level are the same, they **operate on different representations** of the detector and therefore **introduce different information**, leading to different effects.
> >
> > * **Feature-level interaction** injects **CLIP’s global semantics** to help stabilize spatial structure and improve the quality of region features, which in turn benefits the localization capability of the detector head.
> >
> > * **Embedding-level fusion** incorporates **CLIP's image-text alignment** to region embeddings, significantly enhancing their classification ability for novel categories. Moreover, it can also bring improvement in localization, as it refines the scoring of candidate proposals and therefore benefits post-processing in the detection pipeline.
> >
> > To clearly demonstrate each level's influences, we separately evaluate localization and classification metrics, and the results are listed in the following table.
> >
> > * **Localization:** We adopt a binary target/background metric to measure the localization ability (suggested by Reviewer bUuB, which we denote as Loc-AP/AR). The results show clear improvements for both the feature-level and embedding-level fusion, indicating they both contribute to a better localization performance.
> > * **Classification:** We evaluate the performance of using only detector scores on novel categories from two aspects: (1) Top-1 classification accuracy on the ground-truth boxes, and (2) the detection AP50. As the results show, the feature-level interaction brings little improvement for novel-category classification, whereas embedding-level fusion leads to a significant gain, which is consistent with our earlier analysis.
> >
> > Together, these results validate that the two levels play distinct roles and yield different effects: the feature-level fusion primarily improves localization, while the embedding-level fusion primarily enhances classification.
> >
> > We revise the paper to clarify the effect of these two fusions and add the corresponding experiments in Appendix A.2.
> >
> > | Method               | Loc-AP | Loc-AR | Top-1 Acc Novel (Using Only Detection Scores) | AP50 Novel (Using Only Detection Scores) |
> > |----------------------|--------|--------|------------------|------------|
> > | F-ViT                | 25.6   | 44.6   | 0.6              | 5.3        |
> > | F-ViT w/ Feat-level  | 29.8   | 45.9   | 2.2              | 5.0        |
> > | F-ViT w/ Embed-level | 30.6   | 46.2   | 57.2             | 38.5       |
> > | **MCT-Det (ours)**   | 32.0   | 47.1   | 58.3             | 38.7       |

---

### Official Review · Reviewer_wCfi · 2025-11-03

**Soundness:** 3
**Presentation:** 3
**Contribution:** 3
**Rating:** 8
**Confidence:** 3

**Summary:**

This work aims to improve open-vocabulary object detection by better fusing conventional detectors and CLIP. Specifically, a Multi-level CLIP Transfer (MCT-Det) strategy is proposed, which introduces feature level and embedding level interactions between CLIP branch and detector branch, in additon to conventional score level fusion. Experimental results on OV-COCO and OV-LVIS demonstrate the effectiveness of the proposed method.

**Strengths:**

- In Figure 4, the analysis of detection AP and classification accuracy for CLIP and the detector clearly illustrates their respective strengths and weaknesses. Moreover, the analysis of individual modules within the detector clearly reveals which module’s learning is more crucial for novel classes. These empirical studies strongly support the paper’s motivation and method design, providing valuable insights.
- The proposed method achieves impressive performance on OV-COCO and OV-LVIS.
- The paper is clearly written and easy to read.

**Weaknesses:**

- Equation 6 is somewhat unclear. Does it require to know the groundtruth category of predicted bounding box before calculating the final classification score? Moreover, the paper does not explain how the background scores from the CLIP and Det branches are fused.

**Questions:**

See Weaknesses

---

> ### Author Response · Authors · 2025-12-02
> **Response to Reviewer wCfi**
>
> We thank Reviewer wCfi for the effort in reviewing our work and providing positive comments on our **analysis**, **experimental results**, and **presentation**. We address each concern or question below.
>
> > **W1**: Equation 6 is somewhat unclear. Does it require to know the groundtruth category of predicted bounding box before calculating the final classification score? Moreover, the paper does not explain how the background scores from the CLIP and Det branches are fused.
>
> We thank the reviewer for pointing out this concern.
>
> **(1).**
> **The score-level combination does *not* require knowing the ground-truth label of any predicted box.**
> During inference, for each predicted bounding box $i$, we first compute its CLIP scores ${\mathbf{s} _ \mathrm{CLIP}^i}(j)$ and detector scores ${\mathbf{s} _ \mathrm{Det}^i}(j)$ for every candidate category $j \in (C _ B \cup C _ N)$. Then, we combine the two scores via geometric mean using the category-dependent coefficients $\alpha$ and $\beta$:
>
> $$
> \mathbf{s}^i =
> \begin{cases}
> {\mathbf{s} _ {\mathrm{CLIP}} ^ i (j)}^{\alpha} \cdot {\mathbf{s} _ {\mathrm{Det}} ^ i (j)}^{(1-\alpha)} & \text{if } j \in C _ B \\\\
> {\mathbf{s} _ {\mathrm{CLIP}} ^ i (j)}^{\beta} \cdot {\mathbf{s} _ {\mathrm{Det}} ^ i (j)}^{(1-\beta)} & \text{if } j \in C _ N
> \end{cases}
> $$
>
> $\alpha$ and $\beta$ depend only on whether a category $j$ belongs to the base set $C _ B$ or novel set $C _ N$ (i.e., whether it was seen during training). This base/novel split is pre-defined and fixed throughout training and inference.
> The final predicted label is obtained via an $\arg\max$ over the combined scores.
> Thus, the score-level combination does not require any box-level ground-truth information.
>
> **(2).**
> **We treat background as a *base* category and use the same coefficient $\alpha$ as other base classes to fuse its scores.** In our implementation, we construct a CLIP text embedding for the word "background" and include it in the classifier vocabulary during training, so it is naturally treated as a base category.
>
> We update Section 3.5 to explicitly describe the score-level combination and how background scores are handled.

---

### Official Review · Reviewer_E8sp · 2025-11-04

**Soundness:** 2
**Presentation:** 2
**Contribution:** 2
**Rating:** 2
**Confidence:** 4

**Summary:**

This paper proposes MCT-Det, a method that addresses the problem of Open-Vocabulary Object Detection. The method is based on the F-ViT architecture, with the contribution of adding the **feature-level interaction**, which is the cross-attention between the base and the novel branch, and the **embedding level fusion**, which is the fusion of the base branch with the novel branch for score regression. The experiments indicate the efficiency of the proposed method.

**Strengths:**

1. The feature injection from the novel branch to the base branch is interesting, since it can bring the knowledge from the novel category to the base branch for better classification of the base branch.
2. The experiment and ablation studies somehow indicate the efficacy of the proposed components on top of the F-ViT model.

**Weaknesses:**

There are several weaknesses in this paper:
1. Motivation: It seems the score level combination mechanism is one of the motivations and contributions of this paper. However, the geometric mean of the score combination is a well-established method in the field of OVOD [1], [2], [3].

2. Comparison with Large Vision-Language Model: Recent approaches for Large Vision-Language models, including [4], [5], can perform Object Detection for any class based on the prompt provided to the model. A comparison with their approach will help the reader understand why we need OVOD, which can only perform on a single task (Object Detection).

3. Unified head terms: The paper uses the term "unified head"; however, the classification score is the combination of the CLIP score and the classification score from the model, which can be treated as two heads. Also, it does not seem that the unified head can reduce the number of computations. Based on Table 1, the contribution of the Unified head is minor.

4. The contribution of **Feature-Level Interaction** , and **Embedding-Level Fusion** are two simple mechanisms (try to add the interaction of the novel branch to the base branch), most of the components (architecture, training, and inference strategies) rely completely on F-VLM and F-ViT. This weakens the overall strength of the paper’s contribution.

Reference:

[1] Kuo, Weicheng, et al. "F-vlm: Open-vocabulary object detection upon frozen vision and language models." arXiv preprint arXiv:2209.15639 (2022).

[2] Du, Yu, et al. "Learning to prompt for open-vocabulary object detection with vision-language model." Proceedings of the IEEE/CVF conference on computer vision and pattern recognition. 2022.

[3] Pham, Chau, Truong Vu, and Khoi Nguyen. "LP-OVOD: Open-vocabulary object detection by linear probing." Proceedings of the IEEE/CVF Winter Conference on Applications of Computer Vision. 2024.

[4] Rasheed, Hanoona, et al. "Glamm: Pixel grounding large multimodal model." Proceedings of the IEEE/CVF Conference on Computer Vision and Pattern Recognition. 2024.

[5] Zhang, Tao, et al. "Omg-llava: Bridging image-level, object-level, pixel-level reasoning and understanding." Advances in neural information processing systems 37 (2024)

**Questions:**

1. What is the difference in terms of performance between this proposed method for OVOD compared to Large-Vision Language Model?
2. What is the computation cost of this method using the unified heads compared with F-ViT and F-VLM, both of which use two heads?

---

> ### Author Response · Authors · 2025-12-02
> **Response to Reviewer E8sp 1/3**
>
> We thank Reviewer E8sp for the effort in reviewing our work and providing detailed comments. We also appreciate the reviewer's acknowledgement of our idea of **feature-injection** and **experimental results**. We address each concern or question below.
>
> > **W1:** Motivation: It seems the score level combination mechanism is one of the motivations and contributions of this paper. However, the geometric mean of the score combination is a well-established method in the field of OVOD [1], [2], [3].
> >
> > [1] Kuo, Weicheng, et al. "F-vlm: Open-vocabulary object detection upon frozen vision and language models." arXiv preprint arXiv:2209.15639 (2022).
> >
> > [2] Du, Yu, et al. "Learning to prompt for open-vocabulary object detection with vision-language model." Proceedings of the IEEE/CVF conference on computer vision and pattern recognition. 2022.
> >
> > [3] Pham, Chau, Truong Vu, and Khoi Nguyen. "LP-OVOD: Open-vocabulary object detection by linear probing." Proceedings of the IEEE/CVF Winter Conference on Applications of Computer Vision. 2024.
>
> We thank the reviewer for pointing out this concern.
>
> We guess there may be a misunderstanding. **We do *not* regard score-level combination as one of our contributions**. The score-level combination exists in our baseline F-ViT, and our analysis shows that relying solely on the score-level combination is insufficient to leverage the complementary strengths of CLIP and the detector, which motivates us to propose feature- and embedding-level transfers.
>
> As the experiment results show, the proposed feature- and embedding-level transfers both lead to significant improvements in detection precision compared to using only the score-level. **We would like to clarify that the feature- and embedding-level transfers are our main contributions**, while the insufficiency of score-level is the motivation for our design.
>
> We revise Section 1 to make explicit our main contributions and motivation.
>
> | w/ Feat-level | w/ Embed-level | w/ Score-level | AP50_novel on OV-COCO |
> |:-:|:-:|:-:|:-:|
> |  |  | &#10004; | 33.6       |
> | &#10004; |  | &#10004; | 35.2       |
> |  | &#10004; | &#10004; | 41.3       |
> | &#10004; | &#10004; | &#10004; | 42.3       |

---

> > ### Author Response · Authors · 2025-12-02
> > **Response to Reviewer E8sp 2/3**
> >
> > > **W2:** Comparison with Large Vision-Language Model: Recent approaches for Large Vision-Language models, including [4], [5], can perform Object Detection for any class based on the prompt provided to the model. A comparison with their approach will help the reader understand why we need OVOD, which can only perform on a single task (Object Detection).
> > >
> > > **Q1:** What is the difference in terms of performance between this proposed method for OVOD compared to Large-Vision Language Model?
> > >
> > > [4] Rasheed, Hanoona, et al. "Glamm: Pixel grounding large multimodal model." Proceedings of the IEEE/CVF Conference on Computer Vision and Pattern Recognition. 2024.
> > >
> > > [5] Zhang, Tao, et al. "Omg-llava: Bridging image-level, object-level, pixel-level reasoning and understanding." Advances in neural information processing systems 37 (2024)
> >
> > We thank the reviewer for the suggestion of comparing with recent L-VLMs.
> >
> > L-VLMs and OV detectors follow fundamentally different paradigms. L-VLMs excel at multimodal reasoning, they perform detection tasks via text prompts provided by the user. In contrast, OV detectors are specialized for the detection task, predicting bounding-box predictions with high precision and efficiency. **In general, L-VLMs offer broader applicability, but their *detection accuracy* is lower and their *inference speed* is slower compared with OV detectors [1].**
> >
> > The mentioned works (GLaMM [2] and OMG-LLaVA [3]) support only referring or segmentation-style grounding tasks, and cannot be directly evaluated on OVOD benchmarks since they do not produce confidence scores required for detection metrics. To address the reviewer’s concern, we therefore evaluated VLM-R1 [4], a recent representative L-VLM that provided evaluation results on the OVOD task. We compare VLM-R1 with our MCT-Det on the OV-COCO benchmark.
> >
> > As the results show, VLM-R1 achieves comparable performance to our MCT-Det on AP50 novel and AP large, but its overall AP50 is much lower.
> > The most prominent issue is its significantly lower AP small and AR, indicating that current L-VLMs struggle in localizing objects in the whole image, especially for those small objects.
> > In addition, with billion-scale parameters, VLM-R1 shows noticeably higher inference latency (about 3 seconds per image, tested on RTX-3090).
> >
> > We believe that with future advances in model architectures, multimodal modeling, and training paradigms, L-VLMs may eventually become more competitive for detection-oriented applications. Overall, L-VLMs and OV detectors should be viewed as complementary but distinct problem settings.
> >
> > | Method  | Arch. | Params. | AP50_novel | AP50 | AP_small | AP_large | AR@100 | Latency(ms) |
> > |---------|:-:|:-:|-|-|-|-|-|-|
> > | VLM-R1  | Qwen2.5-VL-3B | 3.5B | 41.1 | 31.6 | 4.3 | 38.1 | 31.3 | ~3000 |
> > | **MCT-Det (Ours)** | ViT-B + R-CNN | 108M | 43.1 (+2.0) | 52.0 (+20.4) | 14.8 (+10.5) | 40.9 (+2.8) | 43.9 (+12.6) | 76.4 |
> >
> > > **W3:** Unified head terms: The paper uses the term "unified head"; however, the classification score is the combination of the CLIP score and the classification score from the model, which can be treated as two heads. Also, it does not seem that the unified head can reduce the number of computations. Based on Table 1, the contribution of the Unified head is minor.
> > >
> > > **Q2:** What is the computation cost of this method using the unified heads compared with F-ViT and F-VLM, both of which use two heads?
> >
> > We appreciate the reviewer’s comment.
> >
> > **The "unified head" in our paper does *not* refer to the combination of CLIP and detector scores.** Instead, it refers to using a single detector head to perform both classification and box regression, rather than employing two separate heads as in the standard F-ViT/F-VLM design. The main purpose of this design is to produce localization-aware embeddings, but not to reduce computational cost.
> >
> > To address the reviewer's concern, we measured inference latency for both F-ViT and MCT-Det using separate heads and a unified head on OV-COCO. In both cases, the unified head exhibits a very small reduction in inference latency while consistently improving the detection precision (+0.8 AP50 novel). Although this improvement is smaller than our two main contributions (i.e., feature- and embedding-level fusion), it still provides a measurable benefit and contributes to the overall performance.
> >
> > We add these results in Appendix A.2 to clarify the intended role of the unified head.
> >
> > | Method                     | AP50_novel | Latency(ms) |
> > |----------------------------|:-:|:-:|
> > | F-ViT w/ Separate Heads    | 33.6       | 60.1        |
> > | F-ViT w/ Unified Head      | 34.4       | 60.0        |
> > | MCT-Det w/ Separate Heads  | 42.3       | 76.6        |
> > | **MCT-Det w/ Unified Head (Ours)** | 43.1       | 76.4        |

---

> > > ### Author Response · Authors · 2025-12-02
> > > **Response to Reviewer E8sp 3/3**
> > >
> > > > **W4:** The contribution of Feature-Level Interaction , and Embedding-Level Fusion are two simple mechanisms (try to add the interaction of the novel branch to the base branch), most of the components (architecture, training, and inference strategies) rely completely on F-VLM and F-ViT. This weakens the overall strength of the paper’s contribution.
> > >
> > > We thank the reviewer for the comment.
> > > We would like to clarify that the simplicity of mechanisms should not be equated with limited contribution.
> > >
> > > The proposed feature-level interaction and embedding-level fusion represent a simple and effective instantiation of a more general principle:
> > > **general multimodal representations can be systematically transferred to strengthen the task-specific representations required by downstream applications.**
> > >
> > > In our method, the injected CLIP-derived features into the detector can stabilize RoI representations and further enhance their quality.
> > > As shown in the table below, both the feature- and embedding-level fusion mechanisms exhibit significant improvements on AP50 novel, resulting in a +9.5 performance gain of our MCT-Det over the baseline F-ViT, demonstrating the effectiveness and potential of transferring CLIP knowledge to the OVOD task.
> > >
> > > To the best of our knowledge, MCT-Det is the first work to successfully leverage CLIP features to enhance the detector's representations in the OVOD task. We provide a simple yet effective instantiation of this idea, and we hope this finding will inspire future research on transferring generalized cross-modal representations into specialized recognition tasks.
> > >
> > > | Models               | AP50_novel  |
> > > |----------------------|-|
> > > | F-ViT                | 33.6        |
> > > | F-ViT w/ Feat-level  | 35.2 (+1.6) |
> > > | F-ViT w/ Embed-level | 41.3 (+7.7) |
> > > | **MCT-Det (Ours)**   | 43.1 (+9.5) |
> > >
> > > **References**
> > >
> > > [1] Sapkota, R. and Karkee, M., 2025. Object detection with multimodal large vision-language models: An in-depth review. arXiv preprint arXiv:2508.19294.
> > >
> > > [2] Rasheed, H., Maaz, M., Shaji, S., Shaker, A., Khan, S., Cholakkal, H., Anwer, R.M., Xing, E., Yang, M.H. and Khan, F.S., 2024. Glamm: Pixel grounding large multimodal model. In Proceedings of the IEEE/CVF Conference on Computer Vision and Pattern Recognition (pp. 13009-13018).
> > >
> > > [3] Zhang, T., Li, X., Fei, H., Yuan, H., Wu, S., Ji, S., Loy, C.C. and Yan, S., 2024. Omg-llava: Bridging image-level, object-level, pixel-level reasoning and understanding. Advances in neural information processing systems, 37, pp.71737-71767.
> > >
> > > [4] Shen, H, Liu, P., Li, J., Fang, C., Ma, Y., Liao, J., Shen, Q., Zhang, Z., Zhao, K., Zhang, Q. and Xu, R., 2025. Vlm-r1: A stable and generalizable r1-style large vision-language model. arXiv preprint arXiv:2504.07615.

---

### Official Review · Reviewer_ZTWa · 2025-11-05

**Soundness:** 2
**Presentation:** 3
**Contribution:** 2
**Rating:** 6
**Confidence:** 4

**Summary:**

This study identifies the limitations of the score-level combination in existing open-vocabulary object detection systems (i.e., F-ViT). CLIP assigns similar scores to poor and well-located boxes, while the detector misclassifies novel objects as background. To more effectively leverage the complementary strengths of CLIP and the detector, the authors proposed MCT-Det, a Multi-Level CLIP Transfer strategy that additionally fuses CLIP and multi-level detector features at the feature and embedding levels. In the experiment, MCT-Det achieves remarkable performance gains on the two challenging open-vocabulary detection benchmarks: OV-COCO and OV-LVIS.

**Strengths:**

1. The motivation is clear, and the paper is easy to follow. The idea of exploiting multilevel features and employing earlier feature fusion of CLIP and object detectors is intuitive and straightforward.

2. The overall performance gains on mainstream OV benchmarks are significant, with an improvement of 8.6 mAP on OV-COCO and 5.1 on OV-LVIS.

3. The authors have provided comprehensive ablation studies on the core components of MCT-Det.

**Weaknesses:**

1. This paper hypothesizes the proficiency of object detectors' class-agnostic object localization, as stated in Line 185. It is recommended to provide quantitative support for this hypothesis. Moreover, it is also worth investigating whether the feature-level and embedding-level fusion can also benefit object localization (RPN).

2. The transfer performance of the object detectors on unseen datasets (e.g., objects 365) is absent in the experiment.

3. The authors use the self-distilled CLIP models in the experiment. It would be interesting to see the performance of the original CLIP and more recent SigLIP models.

**Questions:**

1. Is the embedding-level fusion actually an addition of CLIP embeddings and region embeddings? If so, it seems to be equivalent to multiplying scores in the score combination part.

2. Is MCT-Det also applicable to CNN-based frameworks?

---

> ### Author Response · Authors · 2025-12-02
> **Response to Reviewer ZTWa 1/3**
>
> We thank Reviewer ZTWa for the effort in reviewing our work and providing supportive comments. We sincerely appreciate the reviewer’s recognition of our **clear motivation**, and **the effectiveness demonstrated by our experimental results**. We address each concern or question below.
>
>
> > **W1:** This paper hypothesizes the proficiency of object detectors' class-agnostic object localization, as stated in Line 185. It is recommended to provide quantitative support for this hypothesis. Moreover, it is also worth investigating whether the feature-level and embedding-level fusion can also benefit object localization (RPN).
>
> We thank the reviewer for raising this point.
>
> **(1).**
> To quantitatively assess the class-agnostic localization capability of the detector (i.e., whether the detector can localize novel categories when only trained on base categories), we evaluated the average recall on novel categories of our MCT-Det on OV-COCO. In addition, we also directly measure the detector's localization ability via a binary target/background metric (suggested by Reviewer bUuB, which we denote as Loc-AP/AR).
>
> The results show that compared to the novel-trained detector, the base-trained detector still maintains a high recall on novel objects (72.1 vs 80.1, ~10% drop), while its AP suffers a much larger decline (43.1 vs 62.7, ~30% drop). Meanwhile, the improvements in Loc-AP (32.0 vs 33.6) and Loc-AR (47.1 vs 48.4) metrics are relatively small, indicating that the detector already acquires sufficiently class-agnostic localization ability from base-category training. These observations support the hypothesis stated in the paper that the detector provides reliable class-agnostic localization ability.
>
> We included these results in Appendix A.2.
>
> | Training Data | AP50_novel | AR50_novel@100 | Loc-AP | Loc-AR |
> |---------------|:-:|:-:|:-:|:-:|
> | Base          | 43.1 | 72.1 | 32.0 | 47.1 |
> | Base+Novel    | 62.7 | 80.1 | 33.6 | 48.4 |
>
> **(2).**
> **We do *not* expect feature- and embedding-level fusion can benefit RPN.** The proposed feature- and embedding-level fusion focus on enhancing RoI features and the detector head and do not involve the RPN pipeline. Moreover, the current fusion mechanisms are not directly applicable to RPN or earlier stages, and adapting them would require architectural changes that are beyond the scope of the rebuttal period.
>
> We agree that exploring how CLIP features might benefit earlier localization stages (RPN) is an interesting research direction. We appreciate this insightful suggestion and will leave this part as future work for exploration.
>
>
> > **W2:** The transfer performance of the object detectors on unseen datasets (e.g., objects 365) is absent in the experiment.
>
> We thank the reviewer for pointing out this concern.
>
> The transfer performance on unseen datasets (i.e., the detector is trained on OV-LVIS and evaluated on COCO and Objects365) **was *originally* included in Appendix A.4**. These results demonstrate the generalization ability and robustness of our MCT-Det across diverse datasets. To make this clearer, we move the results to Section 4.4 in the main paper.
>
> | Method                     | COCO AP | COCO AP50 | COCO AP75 | Objects365 AP | Objects365 AP50 | Objects365 AP75 |
> |----------------------------|:-:|:-:|:-:|:-:|:-:|:-:|
> | ViLD [1]                   | 36.6    | 55.6      | 39.8      | 11.8           | 18.2             | 12.6             |
> | DetPro [2]                 | 34.9    | 53.8      | 37.4      | 12.1           | 18.8             | 12.9             |
> | BARON-KD [3]               | 36.2    | 55.7      | 39.1      | 13.6           | 21.0             | 14.5             |
> | F-VLM [4]                  | 39.8    | 61.6      | 43.8      | 17.7           | 27.4             | 19.1             |
> | OV-DQUO [5]                | 39.2    | -         | 42.5      | 18.4           | -                | 19.6             |
> | F-ViT + CLIPSelf [6]       | 40.5    | 63.8      | **44.3**      | 19.5           | 31.3             | 20.7             |
> | **MCT-Det + CLIPSelf (Ours)**  | **40.6**| **64.3**  | 43.5      | **21.4**       | **33.9**         | **22.9**         |

---

> > ### Author Response · Authors · 2025-12-02
> > **Response to Reviewer ZTWa 2/3**
> >
> > > **W3:** The authors use the self-distilled CLIP models in the experiment. It would be interesting to see the performance of the original CLIP and more recent SigLIP models.
> > >
> > > **Q2:** Is MCT-Det also applicable to CNN-based frameworks?
> >
> > We thank the reviewer for these valuable suggestions.
> >
> > **MCT-Det is compatible with both CNN-based and ViT-based CLIP backbones, as well as SigLIP [7] or other CLIP variants.**
> >
> > To address the reviewer's concern, we evaluate MCT-Det with different CLIP backbones on OV-COCO, including the original ViT-based CLIP, SigLIP, and CNN-based CLIP.
> > Across all these backbones, MCT-Det consistently yields significant improvements over the corresponding baselines, demonstrating its backbone-agnostic effectiveness and applicability.
> >
> > We include these results in Appendix A.2 to demonstrate the applicability to various backbone of our method.
> >
> > | Method            | Backbone                 | AP50_novel    | AP50_base | AP50 |
> > |-------------------|---------------------------|---------------|:-:|:-:|
> > | F-ViT             | OpenAI-CLIP ViT-B/16      | 16.0          | 36.9      | 31.4 |
> > | **MCT-Det (ours)**| OpenAI-CLIP ViT-B/16      | 23.5 (+7.5)   | 41.0      | 36.4 |
> > | F-ViT             | SigLIP ViT-B/16           | 16.3          | 40.6      | 34.2 |
> > | **MCT-Det (ours)**| SigLIP ViT-B/16           | 23.0 (+6.7)   | 42.2      | 37.2 |
> > | F-ViT             | EVA-CLIP ViT-B/16         | 17.5          | 41.0      | 34.9 |
> > | **MCT-Det (ours)**| EVA-CLIP ViT-B/16         | 26.9 (+9.4)   | 43.8      | 39.4 |
> > | F-ViT             | OpenAI-CLIP RN50          | 24.1          | 44.9      | 39.5 |
> > | **MCT-Det (ours)**| OpenAI-CLIP RN50          | 30.5 (+6.4)   | 46.2      | 42.1 |
> > | F-ViT             | OpenAI-CLIP RN50x4        | 29.4          | 49.9      | 44.6 |
> > | **MCT-Det (ours)**| OpenAI-CLIP RN50x4        | 34.6 (+5.2)   | 50.4      | 46.3 |
> >
> > > **Q1:** Is the embedding-level fusion actually an addition of CLIP embeddings and region embeddings? If so, it seems to be equivalent to multiplying scores in the score combination part.
> >
> > We thank the reviewer for the valuable question. The embedding-level fusion is an **normalized addition** of CLIP and detector's region embeddings, and therefore it **does *not* equal to multiplying CLIP and detector scores.** The detailed explanation is explained below.
> >
> > The goal of embedding-level fusion is to combine CLIP’s multimodal alignment with localization awareness. As we implement this via **normalized addition**, the mutual relationship between two embeddings appears as an **exponential term** in the final scores. To illustrate this more concretely, we present the mathematical derivation to clarify how the final scores are related to the embeddings.
> >
> > For each proposal $i$, the embedding-level fusion is not a direct addition of the CLIP embedding $\mathbf{e} _ {\mathrm{CLIP}}^i$ and the detector embedding $\mathbf{e} _ {\mathrm{Det}}^i$, instead, it is implemented by adding the normalized embeddings,
> >
> > $$
> > \tilde{\mathbf{e}} _ {\mathrm{Det}}^{i}
> > = \mathrm{Norm}(\mathbf{e} _ {\mathrm{CLIP}}^{i}) + \mathrm{Norm}(\mathbf{e} _ {\mathrm{Det}}^{i})
> > = \frac{\mathbf{e} _ {\mathrm{CLIP}}^{i}}{\|\mathbf{e} _ {\mathrm{CLIP}}^{i}\|} +
> > \frac{\mathbf{e} _ {\mathrm{Det}}^{i}}{\|\mathbf{e} _ {\mathrm{Det}}^{i}\|}
> > . \tag 1
> > $$
> >
> > We then calculate the cosine similarity between CLIP embeddings and text embeddings $\mathbf{t}$:
> >
> > $$
> > \cos(\mathbf{e} _ {\mathrm{CLIP}}^{i}, \mathbf{t})
> > = \frac{\mathbf{e} _ {\mathrm{CLIP}}^{i}\cdot\mathbf{t}}
> > {\|\mathbf{e} _ {\mathrm{CLIP}}^{i}\|\cdot\|\mathbf{t}\|}
> > , \tag 2
> > $$
> >
> > Similarly, the cosine similarity between fused detector embeddings and text embeddings becomes
> > $$
> > \mathrm{cos}(\tilde{\mathbf{e}} _ {\mathrm{Det}}^i, \mathbf{t})
> > = \frac{\tilde{\mathbf{e}} _ {\mathrm{Det}}^{i}\cdot\mathbf{t}}{\|\tilde{\mathbf{e}} _ {\mathrm{Det}}^{i}\|\cdot\|\mathbf{t}\|}
> > = \frac{\cos(\mathbf{e} _ {\mathrm{CLIP}}^{i}, \mathbf{t}) + \mathrm{cos}(\mathbf{e} _ {\mathrm{Det}}^i, \mathbf{t})}
> > {\sqrt{2(1+\rho^{i})}}. \tag 3
> > $$
> > where $\rho^{i}=\mathrm{cos}(\mathbf{e} _ {\mathrm{CLIP}}^i, \mathbf{e} _ {\mathrm{Det}}^i)$.

---

> > > ### Author Response · Authors · 2025-12-02
> > > **Response to Reviewer ZTWa 3/3**
> > >
> > > With temperature $T _ {\mathrm{CLIP}}$, we obtain the CLIP scores
> > >
> > > $$
> > > \mathbf{s} _ {\mathrm{CLIP}}^{i}(j) =
> > > \frac{
> > > \exp\left(
> > > \cos\left(\mathbf{e} _ {\mathrm{CLIP}}^{i}, \mathbf{t} _ {j}\right) / T _ {\mathrm{CLIP}}
> > > \right)
> > > }{
> > > \displaystyle
> > > \sum _ {k\in C}
> > > \exp\left(
> > > \cos\left(\mathbf{e} _ {\mathrm{CLIP}}^{i}, \mathbf{t} _ {k}\right) / T _ {\mathrm{CLIP}}
> > > \right)
> > > } =
> > > \frac{
> > > \exp\left(
> > > \cos\left(\mathbf{e} _ {\mathrm{CLIP}}^{i}, \mathbf{t} _ {j}\right) / T _ {\mathrm{CLIP}}
> > > \right)
> > > }{Z _ {\mathrm{CLIP}}^{i}}, \tag 4
> > > $$
> > >
> > > where
> > >
> > > $$
> > > Z _ {\mathrm{CLIP}}^{i} =
> > > \sum _ {k\in C}
> > > \exp\left(
> > > \cos\left(\mathbf{e} _ {\mathrm{CLIP}}^{i}, \mathbf{t} _ {k}\right) / T _ {\mathrm{CLIP}}
> > > \right). \tag 5
> > > $$
> > >
> > > Similar to CLIP scores, detector scores can be obtained with temperature $T _ {\mathrm{Det}}$:
> > >
> > > $$
> > > \mathbf{s} _ {\mathrm{Det}}^{i}(j) =
> > > \frac{
> > > \exp\left(
> > > \cos\left(\tilde{\mathbf{e}} _ {\mathrm{Det}}^{i}, \mathbf{t} _ {j}\right) / T _ {\mathrm{Det}}
> > > \right)
> > > }{
> > > \displaystyle
> > > \sum _ {k\in C}
> > > \exp\left(
> > > \cos\left(\tilde{\mathbf{e}} _ {\mathrm{Det}}^{i}, \mathbf{t} _ {k}\right) / T _ {\mathrm{Det}}
> > > \right)
> > > } =
> > > \frac{
> > > \exp\left(
> > > \cos\left(\tilde{\mathbf{e}} _ {\mathrm{Det}}^{i}, \mathbf{t} _ {j}\right) / T _ {\mathrm{Det}}
> > > \right)
> > > }{Z _ {\mathrm{Det}}^{i}}, \tag 6
> > > $$
> > >
> > > where
> > >
> > > $$
> > > Z _ {\mathrm{Det}}^{i} =
> > > \sum _ {k\in C}
> > > \exp\left(
> > > \cos\left(\tilde{\mathbf{e}} _ {\mathrm{Det}}^{i}, \mathbf{t} _ {k}\right) / T _ {\mathrm{Det}}
> > > \right) =
> > > \sum _ {k\in C}
> > > \exp\left(
> > > \frac{
> > > \cos\left(\mathbf{e} _ {\mathrm{CLIP}}^{i}, \mathbf{t} _ {k}\right) + \cos\left(\mathbf{e} _ {\mathrm{Det}}^{i}, \mathbf{t} _ {k}\right)}
> > > {T _ {\mathrm{Det}} \sqrt{2(1+\rho^{i})}}
> > > \right). \tag 7
> > > $$
> > >
> > > The final score is a combination of two scores via geometric mean:
> > >
> > > $$
> > > \begin{aligned}
> > >   \mathbf{s}^{i}(j) &=
> > >   \big(\mathbf{s} _ {\mathrm{CLIP}}^{i}(j)\big)^{p}
> > >   \cdot
> > >   \big(\mathbf{s} _ {\mathrm{Det}}^{i}(j)\big)^{1-p} \\\\
> > >   &= \frac{
> > >   \exp\left(
> > >   \Big(\frac{p}{T _ {CLIP}}+\frac{1-p}{T _ {Det} \sqrt{2(1+\rho^{i})}}\Big)
> > >   \cos\left(\mathbf{e} _ {\mathrm{CLIP}}^{i}, \mathbf{t} _ {j}\right)
> > >   \right)
> > >   }{
> > >   {Z _ {\mathrm{CLIP}}^{i}}^{p}
> > >   }
> > >   \cdot
> > >   \frac{
> > >   \exp\left(
> > >   \frac{1-p}{T _ {Det} \sqrt{2(1+\rho^{i})}}
> > >   \cos\left({\mathbf{e}} _ {\mathrm{Det}}^{i}, \mathbf{t} _ {j}\right)
> > >   \right)
> > >   }{
> > >   {Z _ {\mathrm{Det}}^{i}}^{1-p}
> > >   }.
> > > \end{aligned} \tag 8
> > > $$
> > >
> > > We denote the two coefficients as $a^i$ and $b^i$:
> > >
> > > $$
> > > a^{i} := \frac{p}{T _ {CLIP}} + \frac{1-p}{T _ {Det} \sqrt{2(1+\rho^{i})}},
> > > \qquad
> > > b^{i} := \frac{1-p}{T _ {Det} \sqrt{2(1+\rho^{i})}}. \tag 9
> > > $$
> > >
> > > And the final scores can be reformulate as:
> > >
> > > $$
> > > \mathbf{s}^{i}(j) =
> > > \frac{
> > > \Big[
> > > \exp\big(
> > > \cos(\colorbox{#ffff00}{$\mathbf{e} _ {\mathrm{CLIP}}^{i}$}, \mathbf{t} _ {j})
> > > \big)
> > > \Big]^{a^{i}}
> > > }{
> > > {Z _ {\mathrm{CLIP}}^{i}}^{p}
> > > }
> > > \cdot
> > > \frac{
> > > \Big[
> > > \exp\big(
> > > \cos(\colorbox{#ffff00}{$\mathbf{e} _ {\mathrm{Det}}^{i}$}, \mathbf{t} _ {j})
> > > \big)
> > > \Big]^{b^{i}}
> > > }{
> > > {Z _ {\mathrm{Det}}^{i}}^{1-p}
> > > }. \tag {10}
> > > $$
> > >
> > > As the equations show, the mutual relationship between the two embeddings appears in the **exponential term**, therefore, final scores cannot be obtained by multiplying or linearly combining the CLIP and detector scores.
> > >
> > >
> > > **References**
> > >
> > > [1] Gu, X., Lin, T.Y., Kuo, W. and Cui, Y., 2021. Open-vocabulary object detection via vision and language knowledge distillation. arXiv preprint arXiv:2104.13921.
> > >
> > > [2] Du, Y., Wei, F., Zhang, Z., Shi, M., Gao, Y. and Li, G., 2022. Learning to prompt for open-vocabulary object detection with vision-language model. In Proceedings of the IEEE/CVF conference on computer vision and pattern recognition (pp. 14084-14093).
> > >
> > > [3] Wu, S., Zhang, W., Jin, S., Liu, W. and Loy, C.C., 2023. Aligning bag of regions for open-vocabulary object detection. In Proceedings of the IEEE/CVF conference on computer vision and pattern recognition (pp. 15254-15264).
> > >
> > > [4] Kuo, W., Cui, Y., Gu, X., Piergiovanni, A.J. and Angelova, A., 2022. F-vlm: Open-vocabulary object detection upon frozen vision and language models. arXiv preprint arXiv:2209.15639.
> > >
> > > [5] Wang, J., Chen, B., Kang, B., Li, Y., Xian, W., Chen, Y. and Xu, Y., 2025, April. Ov-dquo: Open-vocabulary detr with denoising text query training and open-world unknown objects supervision. In Proceedings of the AAAI Conference on Artificial Intelligence (Vol. 39, No. 7, pp. 7762-7770).
> > >
> > > [6] Wu, S., Zhang, W., Xu, L., Jin, S., Li, X., Liu, W. and Loy, C.C., 2023. Clipself: Vision transformer distills itself for open-vocabulary dense prediction. arXiv preprint arXiv:2310.01403.
> > >
> > > [7] Zhai, X., Mustafa, B., Kolesnikov, A. and Beyer, L., 2023. Sigmoid loss for language image pre-training. In Proceedings of the IEEE/CVF international conference on computer vision (pp. 11975-11986).

---

### Author Response · Authors · 2025-12-03
**Summary Comment**

Dear AC and reviewers:

We thank all reviewers for their valuable suggestions and efforts. We also understand the situation following the recent information leak incident and sincerely appreciate the AC for the extra time and workload under these unusual circumstances.

To help ease the AC’s burden, we provide a brief summary of the key points raised by the reviewers.

**We appreciate all the reviewers for acknowledging the strengths of our work:**

* **Clear motivation and empirical analysis:** Reviewers **ZTWa**, **aQPQ**, and **bUuB** remarked that `the motivation is clear`, and reviewer **wCfi** praised our empirical studies `strongly support the paper's motivation and method design, providing valuable insights`.
* **Well-designed method:** Reviewer **ZTWa** noted that the proposed idea is `intuitive and straightforward`, while reviewer **E8sp** considered the design `interesting`.
* **Significant performance and contribution:** Reviewers **ZTWa** and **wCfi** highlighted the impressive performance on OV-COCO and OV-LVIS benchmarks, and reviewer **bUuB** noted that the paper `addresses a key problem` and `makes a significant contribution to this field`.
* **Comprehensive ablation studies:** Reviewer **ZTWa** noted that the ablations cover `the core components`, and reviewer **E8sp** observed that they `somehow indicate the efficacy of the proposed components`. Reviewer **aQPQ** further highlighted that our method `is scalable with the size of the backbone` and `is not sensitive to the hyperparameters`.

**We conducted experiments and provided analysis to address each reviewer's concern as follows:**

* **Localization ability [ZTWa, aQPQ, bUuB]:** We provide quantitative evaluations to demonstrate the class-agnostic localization ability of the detector, and further validate the respective contributions of proposed feature- and embedding-level transfers.
* **Compatibility with more backbones [ZTWa, bUuB]:** We conduct experiments with more CLIP backbones, and the results consistently demonstrate the performance improvements brought by MCT-Det.
* **Unified head [E8sp]:** We evaluate the performance gains and the reduction in latency achieved by adopting the unified head.
* **Comparison with L-VLMs [E8sp]:** We compare MCT-Det with recent L-VLMs to demonstrate its advantages in achieving higher detection accuracy and faster inference.
* **Explanations:** We provide detailed explanations to address each reviewer's concerns, including the question of embedding- and score-level transfers **[ZTWa, wCfi]**, our motivation and contribution **[E8sp, aQPQ]**, comparison with previous works **[aQPQ]**, and concerns about fair comparison **[bUuB]**.

For complete explanations and supporting experiments, please refer to the individual responses below.

**We have revised our main paper and appendix, a new version has been uploaded with all changes highlighted in **yellow** for better visibility:**

* **Title:** We have revised the title from ***Multi-Level CLIP Transfer for Open-Vocabulary Object Detection*** to ***Multi-Level CLIP Knowledge Transfer for Open-Vocabulary Object Detection***.
* **Section 1:** We emphasize our motivation and main contributions.
* **Section 2:** We clarify how our method differs from previous CLIP-based OVOD approaches.
* **Section 3.4:** We explain the role and effect of embedding-level fusion.
* **Section 3.5:** We add the details of the score-level combination.
* **Section 4.5:** We move the cross-dataset evaluation from the appendix to here.
* **Appendix A.2:** We include more ablation experiments, including the ablation on the unified head, using various backbones. We also report the evaluation of the localization ability of the detector.

We sincerely thank the reviewers for their valuable feedback, which has greatly helped us refine this work. We hope this summary provides the AC with a clear understanding of our work and revisions. Thanks again for your time and consideration.

Best regards,

Authors of Submission 11958

---

### Meta-Review · Area_Chair_WAxU · 2026-01-03

**Summary:**

The submission receives initial scores of 6, 2, 8, 4, and 6. The AC finds that the idea of multi-level fusion is simple and effective. However, the main concerns remain that the underlying reasons for its improvement in localization are unclear (see Concern 1), it may rely heavily on CLIPSelf to achieve SoTAs (see Concern 2), and the transfer performance to unseen datasets remains limited. Based on the current version, the AC recommends rejection. The AC encourages the authors to revise the paper based on the reviewers’ comments.

**Reviewer Concerns:**

The reviewers raised several concerns, with the key and common issues summarized below.

1. Reviewers ZTWa, E8sp, and aQPQ question whether and why feature-level and embedding-level fusion benefit object localization.  The rebuttal explains that feature-level interaction injects CLIP’s global semantics to assist localization, while embedding-level fusion incorporates CLIP's image-text alignment to support novel-category classification.

   However, the AC finds that this explanation does not fully convince the reviewers. In particular, it is well known that CLIP’s spatial localization ability is weaker than that of dedicated detectors, and the fundamental reason why CLIP features improve localization is not sufficiently clarified (see Concern 2).
2. Reviewers ZTWa and bUuB question whether the method is effective for original CLIP and more recent SigLIP models, rather than being specifically attached to CLIPSelf. The rebuttal shows that the method achieves clear improvements over F-ViT when using CLIP variants.

   However, compared with other sotas using the same backbone, except F-ViT, the performance is still weaker. This suggests that CLIPSelf appears to be an important contributing factor for MCT-Det. In conjunction with Concern 1, this observation indicates that CLIPSelf itself already enhances the spatial localization capability of the original CLIP, which may explain why MCT-Det is more effective when combined with CLIPSelf. At the same time, this also implies that the observed gains may rely heavily on CLIPSelf’s localization advantage, thereby weakening the independent contribution of the proposed MCT-Det method.
3. Reviewers ZTWa and a5bU point out the lack of comparison experiments, such as transfer to unseen datasets and comparisons with VLMs. The rebuttal provides additional experiments and shows that the proposed method outperforms VLMs.

   However, the transfer performance on unseen datasets, especially on COCO, shows only marginal improvements or is comparable to baselines.
4. Reviewers E8sp and aQPQ argue that the use of two simple fusion strategies weakens the overall contribution and makes it incremental. The rebuttal emphasizes that MCT-Det is the first work to successfully leverage CLIP features to enhance detector representations in the OVOD task.

   The AC considers that simplicity itself is not an issue. However, a simpler design requires a clearer and deeper explanation of why it works. Going beyond high-level claims such as enhanced representations or improved localization, a more fundamental analysis would further strengthen the contribution.
5. Other clarification requests and explanations are well addressed in the rebuttal.

**Reviewer Scores:**

The AC considers that most reviewers will likely maintain their original scores. For Reviewer ZTWa, there is a possibility of a score decrease, as the question of why fusion benefits object localization does not appear to be fully addressed. Reviewer wCfi holds the most positive stance overall, and the main concerns relate to clarification of claims, which are well addressed in the rebuttal.

---

### Decision · Program_Chairs · 2026-01-26

Reject